# Glial Draper signaling triggers cross-neuron plasticity in bystander neurons after neuronal cell death in *Drosophila*

Yupu Wang [1,2,3] ✉, Ruiling Zhang[1,2,4], Sihao Huang[5], Parisa Tajalli Tehrani Valverde[1,2,4], Meike Lobb-Rabe[1,2,6], James Ashley[1,2], Lalanti Venkatasubramanian[7] & Robert A. Carrillo [1,2,4,6] ✉

Neuronal cell death and subsequent brain dysfunction are hallmarks of aging and neurodegeneration, but how the nearby healthy neurons (bystanders) respond to the death of their neighbors is not fully understood. In the *Drosophila* larval neuromuscular system, bystander motor neurons can structurally and functionally compensate for the loss of their neighbors by increasing their terminal bouton number and activity. We term this compensation as cross-neuron plasticity, and in this study, we demonstrate that the *Drosophila* engulfment receptor, Draper, and the associated kinase, Shark, are required for cross-neuron plasticity. Overexpression of the Draper-I isoform boosts cross-neuron plasticity, implying that the strength of plasticity correlates with Draper signaling. In addition, we find that functional cross-neuron plasticity can be induced at different developmental stages. Our work uncovers a role for Draper signaling in cross-neuron plasticity and provides insights into how healthy bystander neurons respond to the loss of their neighboring neurons.

One of the most remarkable features of the brain is its plasticity—the capacity of neural circuits and synapses to adapt to experiences or perturbations by modifying their activity and morphology. Many forms of synaptic plasticity have been reported[1], such as long-term potentiation (LTP)[2], long-term depression (LTD)[3], Hebbian plasticity[4], presynaptic homeostatic potentiation (PHP)[5], presynaptic homeostatic depression (PHD)[6], and others. These mechanisms allow individual synapses to alter their composition and activity during development, learning, injury, and disease. However, during aging and neurodegeneration when substantial neuronal cell death occurs, these plasticity paradigms cannot alleviate the functional defects since the synaptic connections are disrupted, and in many cases, the synapses no longer exist[7]. Interestingly, several studies indicate that neuronal injury or death may alter the structural and functional properties of nearby healthy "bystander" neurons[8–11]. This model highlights that bystander neurons may be an overlooked resource to compensate for nervous system defects during neuronal dysfunction and death.

The first report (to our knowledge) that healthy bystander neurons can respond to injury or death of neighboring neurons was nearly a century ago at the vertebrate neuromuscular junction (NMJ)—denervation of a muscle fiber can induce sprouting of nearby motor neurons (MNs) and eventually restore the circuit function[12–14]. A similar phenotype was observed at invertebrate NMJs. In crustaceans, killing the common MN that innervates multiple muscles induced the bystander MN that co-innervates the muscle to increase its NMJ size and quantal content[15]. In leech embryos, physical removal of the S

[1]Department of Molecular Genetics and Cellular Biology, University of Chicago, Chicago, IL 60637, USA. [2]Neuroscience Institute, University of Chicago, Chicago, IL 60637, USA. [3]Howard Hughes Medical Institute, Janelia Research Campus, Ashburn, VA 20147, USA. [4]Committee on Development, Regeneration, and Stem Cell Biology, University of Chicago, Chicago, IL 60637, USA. [5]Program in Biochemistry and Molecular Biophysics, University of Chicago, Chicago, IL 60637, USA. [6]Program in Cell and Molecular Biology, University of Chicago, Chicago, IL 60637, USA. [7]Department of Zoology, University of Cambridge, Cambridge, UK. ✉e-mail: wangy9@janelia.hhmi.org; robertcarrillo@uchicago.edu

interneuron that resides in one ganglion led to compensation from an S interneuron in a nearby ganglion. The healthy S interneuron extended axons into the lesioned ganglion and restored synaptic function[16]. Recent studies at the *Drosophila* larval NMJ reported similar observations using a genetic approach to ablate MNs[8,10,17]. Most *Drosophila* larval muscles are innervated by two excitatory glutamatergic MNs, known as type-I big MN (Ib MN) and type-I small MN (Is MN) based on their terminal bouton size[18]. In each hemisegment, ~29 Ib MNs innervate thirty muscle fibers mostly in a one-to-one manner[19,20], whereas two Is MNs innervate separate groups of muscles and therefore, are known as the common exciters[21]. Genetic ablation of Is MNs led to NMJ expansion and elevated excitatory post-synaptic potential (EPSP) amplitude from healthy bystander Ib MNs[8,10,17]. Four Ib MNs were examined and three of them (MN1-Ib, MN4-Ib and MN6-Ib) showed compensation at differing levels upon Is MN ablation. In addition, ablating Ib MNs (MN1-Ib) did not induce compensation in the corresponding Is MN, suggesting MNs differ in their ability to compensate[10]. These vertebrate and invertebrate studies provide strong support for healthy bystander neurons in restoring synaptic function upon the loss of their neighbors. However, the mechanisms by which these bystander neurons respond to the death of their neighbors are largely unknown. In this study, we referred to plasticity changes induced by neuronal cell death as "cross-neuron plasticity" and explored the underlying mechanisms.

In the cross-neuron plasticity studies described above, the dying neuron and the bystander neurons do not physically contact or synaptically connect with each other, suggesting a third party is likely involved in sensing the injury and spreading the signal. Glial cells are an attractive candidate because of their close association with neurons and essential roles in neural development, synaptic plasticity, and injury responses[22,23]. During development and metamorphosis when substantial axon and dendrite pruning occurs, and during injury induced neuronal degeneration, glial cells detect and engulf neuronal debris through a conserved signaling pathway mediated by an engulfment receptor, MEGF10 and Jedi (vertebrates)[24,25]/CED-1 (*C-elegans*)[26]/Draper (*Drosophila*)[27–29]. Draper contains an ITAM domain found in many mammalian immunoreceptors which can be phosphorylated by Src42a to allow binding of an SH2 domain kinase, Shark[30]. The Draper/Shark complex recruits the glial membrane for engulfment through dCed-6 and Rac1 and activates engulfment gene expression through the dJNK pathway[31–33]. The necessity of the Draper/Shark pathway in glia-mediated clearance of neuronal debris following injury led to an appealing hypothesis that this engulfment pathway may serve as the trigger to initiate cross-neuron plasticity. Importantly, a recent study found that severing sensory axons in the adult wing influenced cargo transport in bystander axons in a Draper-dependent mechanism[9]. However, whether and how the morphology and physiology of these bystander neurons are affected was not studied.

In this study, we utilize the larval neuromuscular system to examine how bystander neurons detect loss of neighboring neurons. First, we genetically ablate Is MNs and find that Draper is required for clearance of the Is MN axon and cell body debris. Next, we examine structural and functional cross-neuron plasticity in healthy bystander Ib MNs and find that the Draper/Shark signaling pathway is required primarily in glial cells. In addition, Draper-I is the specific isoform mediating cross-neuron plasticity and elevating Draper-I expression can boost plasticity. These data provide insights about how neuronal cell death is detected to induce cross-neuron plasticity in the neuromuscular system. To further explore cross-neuron plasticity, we perform age-dependent Is MN ablation and find that functional plasticity, but not structural plasticity, can be induced at all larval stages. Finally, to determine the behavioral consequences of Ib cross-neuron plasticity, we examine larval locomotion and observe elevated crawling speed. Overall, these data support an important role for healthy bystander neurons in detecting and responding to the nearby neuronal loss.

## Results

### Draper is required for debris clearance after Is MN ablation

The *Drosophila* engulfment receptor, Draper, is implicated in axonal debris clearance in several *Drosophila* nerve injury models including axotomy of olfactory and wing sensory neurons[9,28,34]. However, whether Draper is required for the clearance of axonal debris generated by programmed cell death and the subsequent cross-neuron plasticity is not clear. Here, we genetically ablated Is MNs by ectopic expression of the cell death genes head involution defective (*hid*) and reaper (*rpr*) in a *draper* mutant background (*draper*[Δ5]) and examined Is MN debris clearance. We used a Is MN specific driver, *A8-GAL4* (hereafter named *Is-GAL4*), to label and ablate Is MNs. In a prior study, we demonstrated that expression of *Is-GAL4* begins at embryonic stage 15 and efficiently ablated Is MNs by early first instar stage[8]. In *Is > GFP* larvae, GFP labeled the Is MN axons (Fig. 1a), and co-expression of *rpr,hid* (*Is > GFP,rpr,hid*) led to ablation of Is MNs and complete removal of the Is MN axons (Fig. 1b). However, ablation of Is MNs in *draper* mutant animals led to significant accumulation of GFP in the segmental nerve that co-localized with the glial cell marker, Repo (Fig. 1c–e), suggesting a failure of axonal debris clearance. Next, we examined the ventral nerve cord (VNC) where Is MN cell bodies and dendrites are located. We found complete clearance of Is MN debris in the VNC in *Is > GFP,rpr,hid* first instar larvae (Fig. 1f, g); however, ablation in a *draper* mutant background resulted in significant GFP retention (Fig. 1h-j), similar to the accumulation in the segmental nerve. In summary, we showed that Draper is required to efficiently remove neuronal debris induced by programmed cell death, in both the segmental nerve and the VNC.

### Draper is required for cross-neuron plasticity

We reasoned that clearance of the MN debris may be part of the signaling pathway that initiates cross-neuron plasticity. Therefore, we examined the role of Draper in cross-neuron plasticity. We genetically ablated Is MNs and examined a specific bystander Ib MN that innervates the dorsal muscle 4 (MN4-Ib), because in a previous study, this MN displayed robust structural and functional plasticity when the adjacent Is MN was ablated[8]. We first examined the NMJ size of the MN4-Ib to determine structural plasticity. Ablating Is MNs in a wild type background led to an increase of MN4-Ib bouton number, as previously reported[8] (Fig. 2a, b, e). Interestingly, this NMJ expansion was not observed when Is MNs were ablated in a *draper* mutant background (Fig. 2c–e), suggesting Draper is required for structural plasticity.

Next, we tested the role of Draper in functional cross-neuron plasticity. Muscle 4 receives innervation from both MN4-Ib and the dorsal Is MN, and these neurons are normally activated simultaneously during electrophysiology recordings (Supplementary Fig. 1a). However, to understand the changes of Ib MN activity before and after ablation, we needed to isolate Ib MN activity from a wild type animal where both Ib and Is MNs are present. One approach to separate the activity of these MNs is by tuning the stimulating voltage, as Ib MNs have a lower stimulating threshold than Is MNs[18,35]. Using GCaMP imaging to visualize the activated MN together with electrophysiology recording from the muscle (Supplementary Fig. 1a), we recorded a smaller EPSP which was generated by stimulation of the Ib MN alone, and a larger EPSP which was generated by activation of both Ib and Is MNs (Supplementary Fig. 1b). We normalized the Ib EPSP to Ib+Is EPSP and found that MN4-Ib contributes approximately 56% to the total EPSP, similar to our previous observation[8]. We therefore used this ratio (Ib/Ib+Is) to indicate the MN4-Ib baseline activity. Next, we recorded spontaneous and evoked EPSPs in wild type and *draper* mutant animals, with or without Is MN ablation (Fig. 2f, g). We did not observe any significant changes with spontaneous release in the *draper* mutant background as measured by frequency or amplitude of the spontaneous EPSP (also known as miniature EPSP, mEPSP) (Supplementary Fig. 2). Examination of evoked activity revealed significantly smaller Ib

EPSPs and quantal content when ablating Is MNs in a *draper* mutant background compared to control Is ablated animals (Fig. 2h, i). To better illustrate the data, we normalized the Is ablated EPSP and quantal content to respective controls (i.e. Is ablated EPSP was normalized to control non-ablated EPSP), and compared the normalized data across genotypes, together with the Ib baseline activity (Ib/Ib+Is). We found that upon Is MN ablation, the Ib EPSP was significantly higher than Ib baseline activity, suggesting a robust functional cross-neuron plasticity (Fig. 2j, comparing Ctrl to Ib/Ib+Is). However, this compensation was absent in *draper* mutant animals (Fig. 2j, comparing *drpr*[Δ5] to Ib/Ib+Is). In addition, direct comparison of the normalized EPSP and quantal content confirmed a loss of functional cross-neuron plasticity in the mutant background (Fig. 2j, k). Taken together, our data suggested that Draper is required for both structural and functional cross-neuron plasticity.

## Draper is primarily required in glial cells for cross-neuron plasticity

Draper is expressed in multiple cell types, including glial cells and muscles, where it functions as an engulfment receptor[36,37]. MNs interact extensively with glial cells in both the VNC and segmental nerve bundles as well as with muscles at the NMJ. Therefore, we sought to determine where Draper function is required for cross-neuron

plasticity. Here, we first validated a *draper* RNAi line and found that expressing the RNAi using the glial cell driver, *Repo-GAL4*, or the muscle driver, *Mef2-GAL4*, eliminated Draper expression in respective cells (Supplementary Fig. 3).

We then examined cross-neuron plasticity in animals with cell specific *draper* knockdown. Ablating Is MNs in controls led to an increase of Ib bouton number (Fig. 3a, b, j), and this NMJ expansion was blocked by *draper* double knockdown in both glial cells and muscles (Fig. 3c, d, j), similar to the *draper* mutant phenotype (Fig. 2e). *draper* single knockdown in glial cells blocked the elevation of Ib bouton numbers (Fig. 3e, f, j) while the muscle knockdown did not (Fig. 3g, h, j), suggesting that Draper is specifically required in glial cells. However, we noticed an increase of Ib bouton number upon glial *draper* knockdown (Fig. 3j), which might block the further structural compensation upon Is ablation. Next, we performed electrophysiology analyses in these knockdown conditions (Fig. 3i and Supplementary Fig. 4). We found that knocking down *draper* in glial cells fully blocked the Ib EPSP compensation, when comparing the normalized EPSP to control Is ablated animals or to the Ib baseline activity (Fig. 3k). Similarly, the elevated Ib quantal content induced by Is ablation was blocked when knocking down *draper* in glial cells (Fig. 3l). Removing *draper* in muscles also caused a decrease of Ib EPSP compensation (Fig. 3k), but the quantal content was not

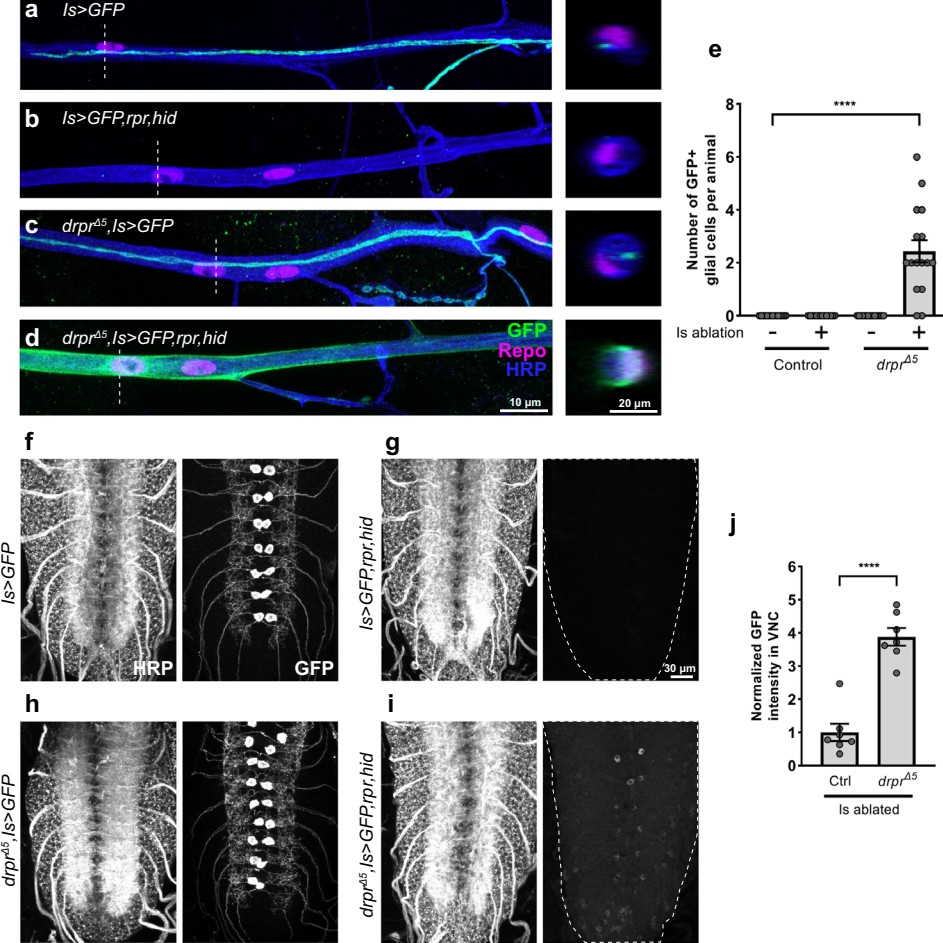

**Fig. 1 | Draper is required for debris clearance after Is MN ablation.** Axon bundles in third instar (**a**) non-ablated control (*Is > GFP*), (**b**) Is ablated (*Is > GFP,hid,rpr*), (**c**) non-ablated *draper* mutant (*drpr*[Δ5], *Is > GFP*) and (**d**) Is ablated *draper* mutant (*drpr*[Δ5], *Is > GFP,hid,rpr*) larvae, labeled with GFP (green), Repo (glial cell marker, magenta) and HRP (neuronal marker, blue). Gray dashed lines indicate the position of cross sections. Significant GFP positive debris accumulated in glial cells when

ablating Is MNs in a *draper* mutant background. **e** Quantification of the number of GFP+ glial cells per animal. F(3,57) = 14.09, p < 0.0001, One-way ANOVA. *N* (larvae) =15, 14, 16, 16. **f–i** VNCs of first instar larvae of displayed genotypes labeled with HRP and GFP. Note the significant amount of GFP signal remaining in (**i**). **j** Quantification of GFP intensity in VNC. t(12) = 7.703, p < 0.0001, unpaired *t* test, two-tailed. N (VNCs) = 7, 7. Error bars indicate ± SEM, ****p < 0.0001.

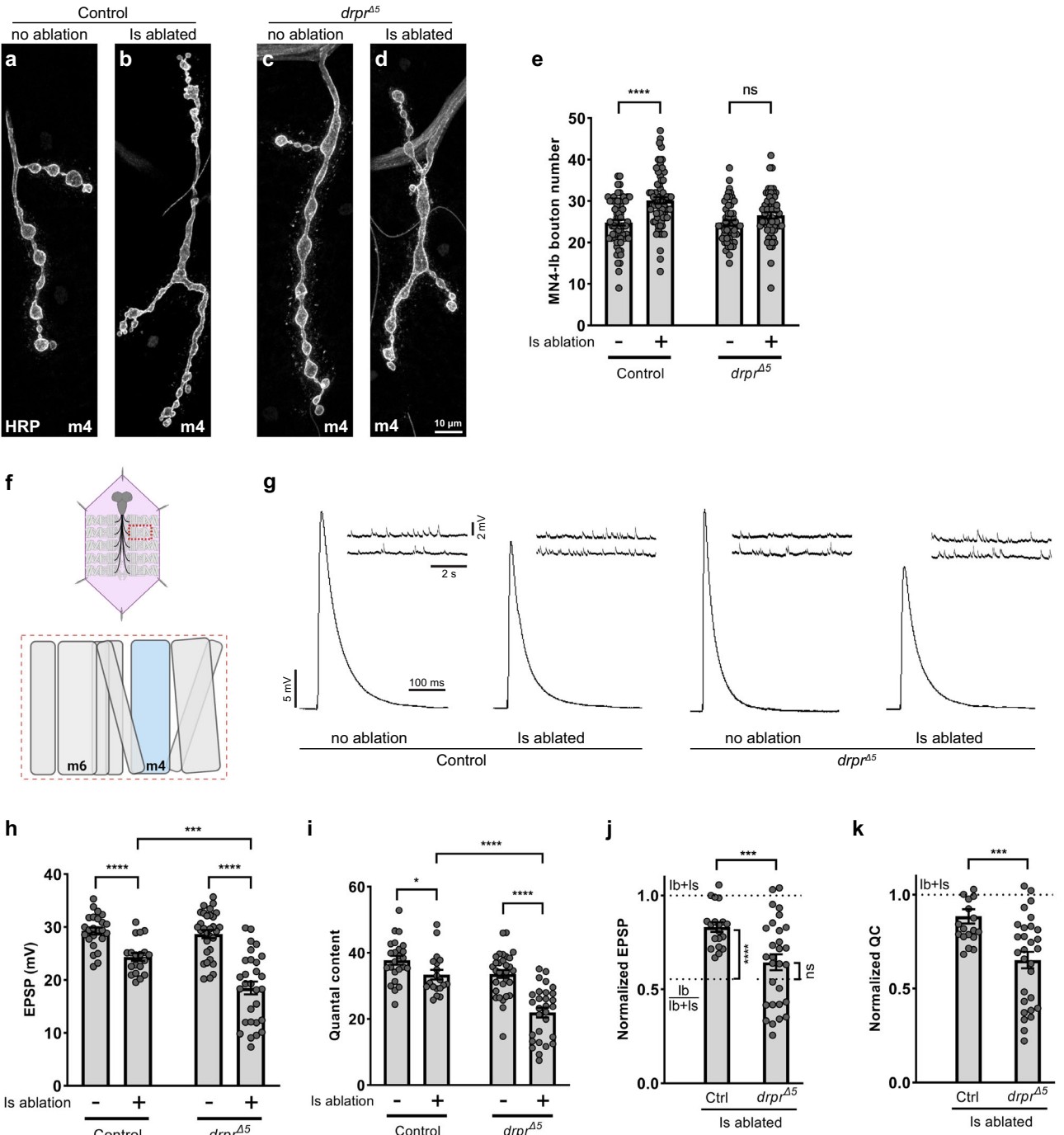

**Fig. 2 | Draper is required for cross-neuron plasticity.** NMJs of MN4-Ib in third instar (**a**) non-ablated control (*Is > GFP*), (**b**) Is ablated (*Is > GFP,hid,rpr*), (**c**) non-ablated *draper* mutant (*drpr^Δ5, Is > GFP*) and (**d**) Is ablated *draper* mutant (*drpr^Δ5, Is > GFP,hid,rpr*) larvae, labeled with HRP (gray). The NMJ was expanded in control Is ablated larvae due to cross-neuron plasticity (**b**), and this expansion is absent in a *draper* mutant background (**d**). **e** Quantification of MN4-Ib bouton numbers in non-ablated and Is ablated larvae in control and *drpr^Δ5* backgrounds. Control (*N* = 66 and 69 NMJs), t(133) = 5.030, *p* < 0.0001, unpaired *t* test, two-tailed. *drpr^Δ5* (*N* = 53 and 56 NMJs), t(107) = 1.838, *p* = 0.0688, unpaired *t* test, two-tailed. **f** Cartoon representation of a dissected larva (pink) and a hemisegment highlighted by dashed red rectangle. The target muscle examined in this figure is shown in blue. Cartoon is generated with Biorender. **g** EPSP and mEPSP traces from non-ablated and Is ablated larvae in control and *drpr^Δ5* backgrounds. **h** Quantification of EPSP amplitude of non-ablated and Is ablated larvae in control and *drpr^Δ5* backgrounds. Control, t(41) = 4.924, *p* < 0.0001, unpaired *t* test, two-tailed. *drpr^Δ5*, t(48.04) = 7.011, *p* < 0.0001, unpaired *t* test, two-tailed, with Welch's correction. Is ablated control vs

Is ablated in *drpr^Δ5*, t(43.54) = 4.075, *p* = 0.0002, unpaired *t* test, two-tailed, with Welch's correction. **i** Quantification of quantal content of non-ablated and Is ablated larvae in control and *drpr^Δ5* backgrounds. Control, t(41) = 2.224, *p* = 0.0317, unpaired *t* test, two-tailed. *drpr^Δ5*, t(58) = 6.194, *p* < 0.0001, unpaired *t* test, two-tailed. Is ablated control vs Is ablated in *drpr^Δ5*, t(46) = 5.304, *p* < 0.0001, unpaired *t* test, two-tailed. For (**h**) and (**i**), *N* (NMJs) = 24, 19, 31, 29. **j** Quantification of normalized EPSP of Is ablated larvae in control and *drpr^Δ5* backgrounds. Is ablated control vs *drpr^Δ5*, t(43.20) = 3.753, *p* = 0.0005, unpaired *t* test, two-tailed, with Welch's correction. Is ablated control vs Ib/Ib+Is, t(29) = 6.506, *p* < 0.0001, unpaired *t* test, two-tailed. Is ablated in *drpr^Δ5* vs Ib/Ib+Is, t(37.82) = 1.524, *p* = 0.1357, unpaired *t* test, two-tailed, with Welch's correction. **k** Quantification of normalized quantal content of Is ablated larvae in control and *drpr^Δ5* backgrounds. t(46) = 3.730, *p* = 0.0005, unpaired *t* test, two-tailed. For (**j**, **k**), N (NMJs) = 19 and 29. Error bars indicate ± SEM, ns = non-significant, \**p* < 0.05, \*\*\**p* < 0.001, \*\*\*\**p* < 0.0001.

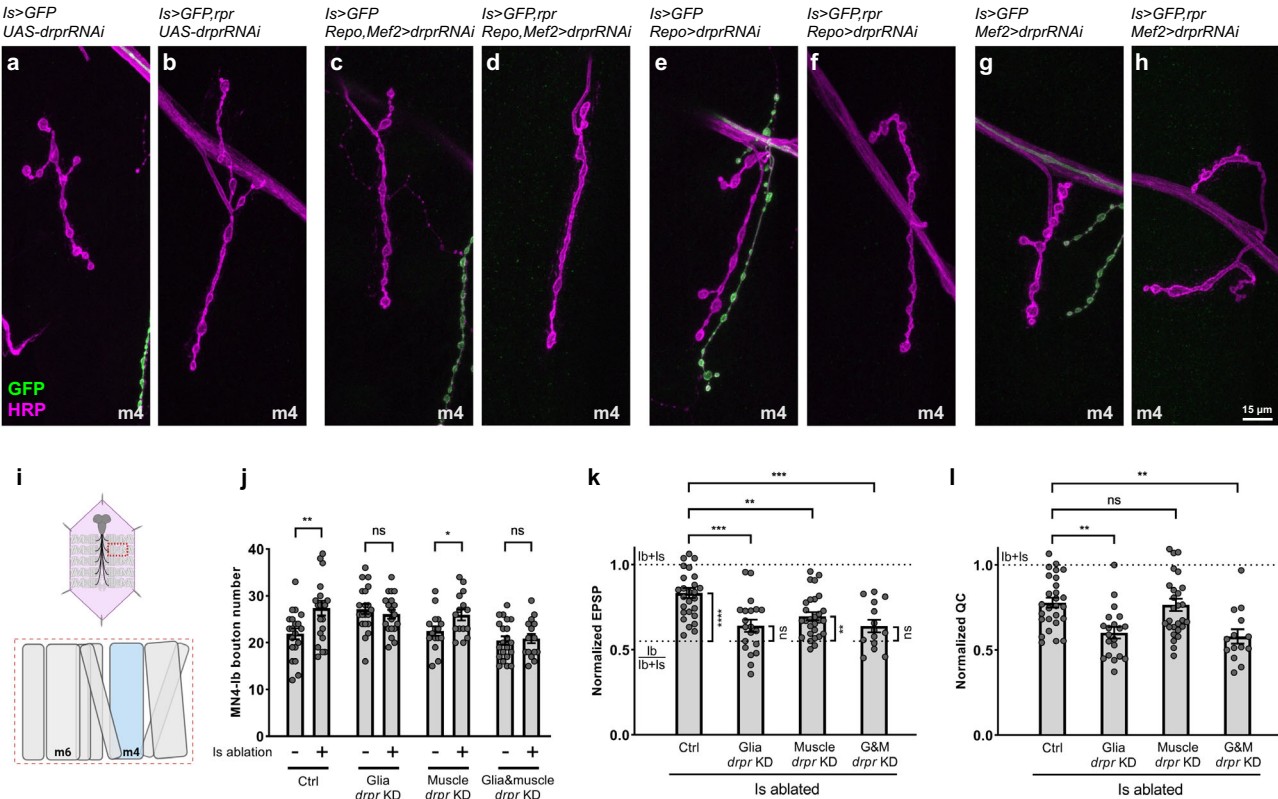

**Fig. 3 | Draper is required in glial cells for cross-neuron plasticity. a–h** NMJs of MN4-Ib in third instar non-ablated and Is ablated larvae in control, glia *draper* knockdown, muscle *draper* knockdown, and double knockdown backgrounds, labeled with GFP (green) and HRP (magenta). NMJ expansion was observed upon Is MN ablation (**b**), and this expansion is absent in double *draper* knockdown (**d**) and glial *draper* knockdown (**h**) backgrounds. **i** Cartoon representation of a dissected larva (pink) and a hemisegment highlighted by dashed red rectangle. The target muscle examined in this figure is shown in blue. Cartoon is generated with Biorender. **j** Quantification of MN4-Ib bouton number between non-ablated and Is Ablated larvae in control, glia *draper* knockdown, muscle *draper* knockdown, and double knockdown backgrounds. Control ($N = 20$ and 21 NMJs), t(39) = 2.822, $p = 0.0075$, unpaired *t* test, two-tailed. Glia *draper* knockdown ($N = 20$ and 19 NMJs), t(37) = 0.7525, $p = 0.4565$, unpaired *t* test, two-tailed. Muscle *draper* knockdown ($N = 16$ and 15 NMJs), t(29) = 2.204, $p = 0.0356$, unpaired *t* test, two-tailed. Double knockdown ($N = 21$ and 16 NMJs), t(35) = 0.2965, $p = 0.7686$, unpaired *t* test, two-tailed. **k** Quantification of normalized EPSP of Is ablated larvae in control, glia

*draper* knockdown, muscle *draper* knockdown, and double knockdown backgrounds. F(3, 85) = 9.191, $p < 0.0001$, One-way ANOVA. Is ablated control vs glia *draper* knockdown, $p = 0.0001$. Is ablated control vs muscle *draper* knockdown, $p = 0.0042$. Is ablated control vs double knockdown, $p = 0.0006$. Is ablated control vs Ib/Ib+Is, t(37) = 5.462, $p < 0.0001$, unpaired *t* test, two-tailed. Is ablated in glia *draper* knockdown vs Ib/Ib+Is, t(30) = 1.483, $p = 0.1486$, unpaired *t* test, two-tailed. Is ablated in muscle *draper* knock down vs Ib/Ib+Is, t(38) = 3.178, $p = 0.0029$, unpaired *t* test, two-tailed. Is ablated in double knockdown vs Ib/Ib+Is, t(24) = 1.540, $p = 0.1367$, unpaired *t* test, two-tailed. **l** Quantification of normalized quantal content of Is ablated larvae in control, glia *draper* knockdown, muscle *draper* knock-down, and double knockdown backgrounds. F(3, 85) = 8.263, $p < 0.0001$, One-way ANOVA. Is ablated control vs glia *draper* knockdown, $p = 0.0028$. Is ablated control vs muscle *draper* knock down, $p = 0.9913$. Is ablated control vs double knockdown, $p = 0.0052$. For (**k**, **l**), (NMJs): 27, 20, 28, 14. Error bars indicate ± SEM, ns = non-significant, $*p < 0.05$, $**p < 0.01$, $***p < 0.001$.

---

significantly different than control Is ablated animals (Fig. 3l), suggesting that Draper is partially required in muscles for cross-neuron plasticity. Finally, *draper* knockdown in both glial cells and muscles fully blocked functional plasticity, similar to knockdown in glial cells alone (Fig. 3k, l). Taken together, we showed that Draper is primarily required in the glial cells to mediate cross-neuron plasticity.

## The Draper co-factor, Shark, is required in cross-neuron plasticity

Previous studies revealed that Draper acts together with an essential kinase, Shark, to regulate engulfment and target gene expression[30,38,39]. Therefore, we hypothesized that Shark might also be required for cross-neuron plasticity. We took an RNAi approach to specifically knock down *shark* in glial cells and muscles. We first assayed for structural NMJ changes and observed that *shark* knockdown in glial cells blocked the Ib structural compensation induced by Is ablation, whereas knockdown in muscles did not (Fig. 4a–f, j). Additionally, *shark* knockdown in both glial cells and muscles blocked Ib structural compensation, similar to glial cell knockdown, suggesting that Shark is

required in glial cells for structural plasticity (Fig. 4g, h, j). Next, we measured the Ib functional compensation in *shark* knockdown animals (Fig. 4i and Supplementary Fig. 5). We found that knocking down *shark* in glial cells led to a significant loss of Ib functional plasticity, as the EPSP and quantal content were both decreased compared to controls (Fig. 4k, l). Muscle knockdown led to a slight decrease of the EPSP, but no significant change of the normalized quantal content (Fig. 4k, l), suggesting that muscles might have a limited role, consistent with our observation in *draper* knockdown experiments (Fig. 3k, l). In addition, simultaneous knockdown of *shark* in both muscles and glia mimicked glial knockdown, further confirming a significant role of *shark* in glial cells in cross-neuron plasticity (Fig. 4j–l). In summary, our data suggested that Draper and its co-factor, Shark, are both required for cross-neuron plasticity.

## Overexpression of Draper-I boosts cross-neuron plasticity of MN6-Ib

After demonstrating that Draper is required for cross-neuron plasticity, we wondered if we could boost plasticity by overexpressing

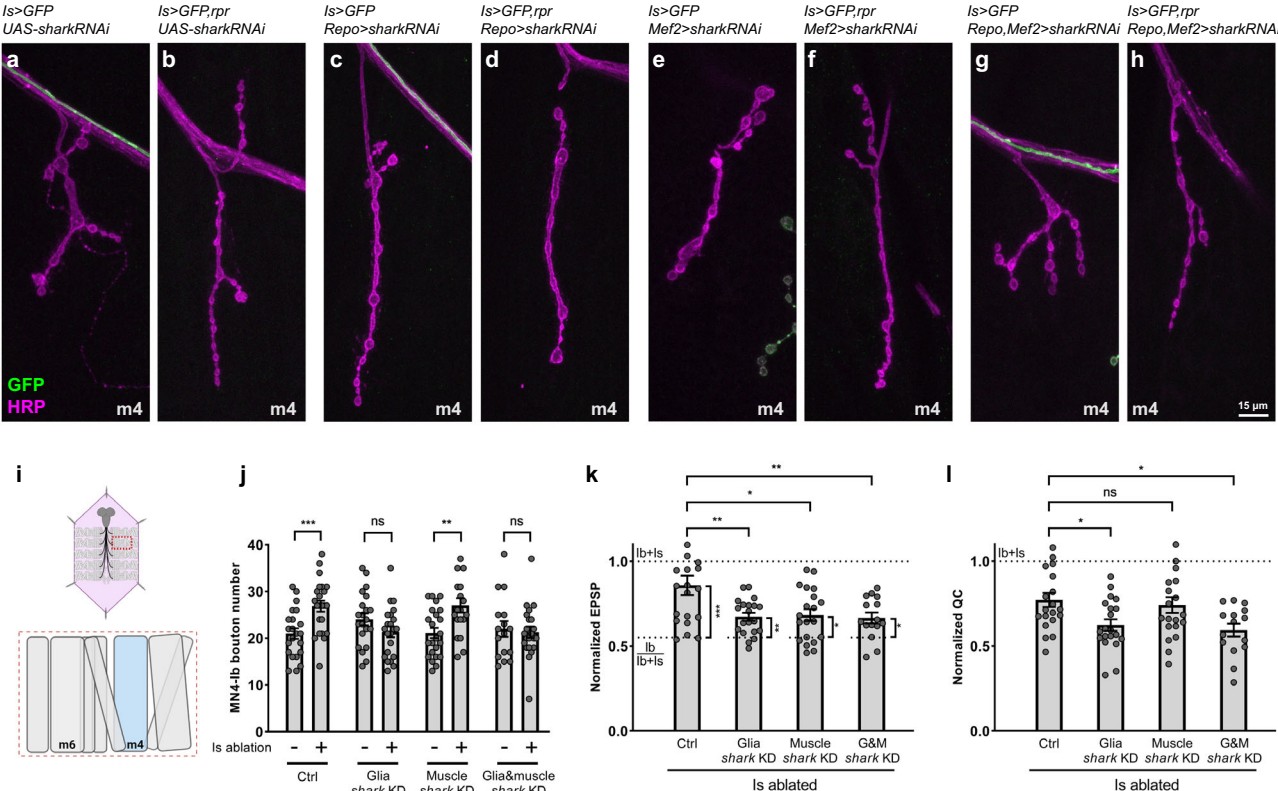

**Fig. 4 | Shark is required in glial cells for cross-neuron plasticity. a–h** NMJs of MN4-Ib in third instar larvae of non-ablated and Is ablated larvae in control, glia *shark* knockdown, muscle *shark* knockdown, and double knockdown backgrounds, labeled with GFP (green) and HRP (magenta). NMJ expansion was observed upon Is MN ablation (**b**), which is blocked by glia *shark* knockdown (**d**) or double knockdown (**h**). **i** Cartoon representation of a dissected larva (pink) and a hemisegment highlighted by dashed red rectangle. The target muscle examined in this figure is shown in blue. Cartoon is generated with Biorender. **j** Quantification of MN4-Ib bouton number in non-ablated and Is ablated larvae in control, glial *shark* knockdown, muscle *shark* knockdown, and double knockdown backgrounds. Control ($N = 22$ and 23 NMJs), t(43) = 3.598, $p = 0.0008$, unpaired *t* test, two-tailed. Glial *shark* knockdown ($N = 21$ and 22 NMJs), t(41) = 1.566, $p = 0.1250$, unpaired *t* test, two-tailed. Muscle *shark* knockdown ($N = 23$ and 19 NMJs), t(40) = 3.220, $p = 0.0025$, unpaired *t* test, two-tailed. Double *shark* knockdown ($N = 16$ and 22 NMJs), t(36) = 0.3390, $p = 0.7366$, unpaired *t* test, two-tailed. **k** Quantification of normalized EPSP of Is ablated larvae in control, glia *shark* knockdown, muscle *shark*

knockdown, and double knockdown backgrounds. F(3, 71) = 5.533, $p = 0.0018$, One-way ANOVA. Is ablated control vs glia *shark* knockdown, $p = 0.0062$. Is ablated control vs muscle *shark* knockdown, $p = 0.0105$. Is ablated control vs double knockdown, $p = 0.0093$. Is ablated control vs Ib/Ib+Is, t(28.04) = 4.485, $p = 0.0001$, unpaired *t* test, two-tailed, with Welch's correction. Is ablated in glia *shark* knockdown vs Ib/Ib+Is, t(30) = 2.798, $p = 0.0089$, unpaired *t* test, two-tailed. Is ablated in muscle *shark* knockdown vs Ib/Ib+Is, t(30) = 2.329, $p = 0.0268$, unpaired *t* test, two-tailed. Is ablated in double knockdown vs Ib/Ib+Is, t(25) = 2.254, $p = 0.0332$, unpaired *t* test, two-tailed. **l** Quantification of normalized quantal content of Is ablated larvae in control, glia *shark* knockdown, muscle *shark* knockdown, and double knockdown backgrounds. F(3, 71) = 4.437, $p = 0.0065$, One-way ANOVA. Is ablated control vs glia *shark* knockdown, $p = 0.0466$. Is ablated control vs muscle *shark* knockdown, $p = 0.9470$. Is ablated control vs double knockdown, $p = 0.0214$. For (**k, l**), N (NMJs) = 20, 20, 20, 15. Error bars indicate ± SEM, ns = non-significant, *$p < 0.05$, **$p < 0.01$, ***$p < 0.001$.

---

*draper*. Draper has three isoforms with distinct functions[37]—Draper-I regulates engulfment of axonal debris through its intracellular immunoreceptor tyrosine-based activation motif (ITAM); Draper-II is an inhibitor of Draper-I, reduces debris clearance when overexpressed, and is selectively expressed in adults; Draper-III lacks the ITAM and its function is unknown. We overexpressed each isoform and examined cross-neuron plasticity.

Overexpressing *draper-I* in either glial cells or muscles did not further increase the bouton number, EPSP or quantal content of MN4-Ib MNs, as they still compensated to a similar level to that observed with Is ablation (Fig. 5a–d, Supplementary Fig. 6). We reasoned that this could be due to a ceiling effect because MN4-Ib MNs already display robust plasticity and thus may be unable to increase plasticity even further. Therefore, we chose to examine MN6-Ib MNs (Fig. 5e), which only showed a modest level of functional cross-neuron plasticity in previous studies[8,17], compared to the significantly higher plasticity at MN4-Ib. In glial or muscle *draper-I* overexpressing animals, MN6-Ib bouton numbers did not show further increase (Fig. 5f and Supplementary Fig. 7a-f). To assay the functional plasticity of MN6-Ib, we established a baseline for MN6-Ib using GCaMP imaging together with

electrophysiology (Supplementary Fig. 1c). Surprisingly, the EPSP amplitude and quantal content significantly increased when over-expressing *draper-I* in either glial cells or muscles, compared to control Is ablated animals or to the baseline Ib/Ib+Is (Fig. 5g, h and Supplementary Fig. 7g–l). These results suggested that overexpressing *draper-I* can boost the Ib functional compensation, but selectively on MN6-Ib. The lack of a further increase of structural compensation could be due to separate structural and functional plasticity mechanism or to a ceiling effect.

Unlike Draper-I, Draper-II is proposed to function as a repressor for debris engulfment[37]. We found that overexpressing *draper-II* in glial cells suppressed the structural compensation as reflected in Ib bouton number, as well as the compensation of EPSP and quantal content on both MN4-Ib (Fig. 6a–d, Supplementary Fig. 8) and MN6-Ib (Fig. 6e–h, Supplementary Fig. 9). Interestingly, overexpressing *draper-II* in muscles suppressed the compensation of Ib bouton numbers of both MNs, but not the EPSP or quantal content. This data fit our previous observation that cross-neuron plasticity is primarily regulated by Draper activity in glial cells, but muscles also partially contribute.

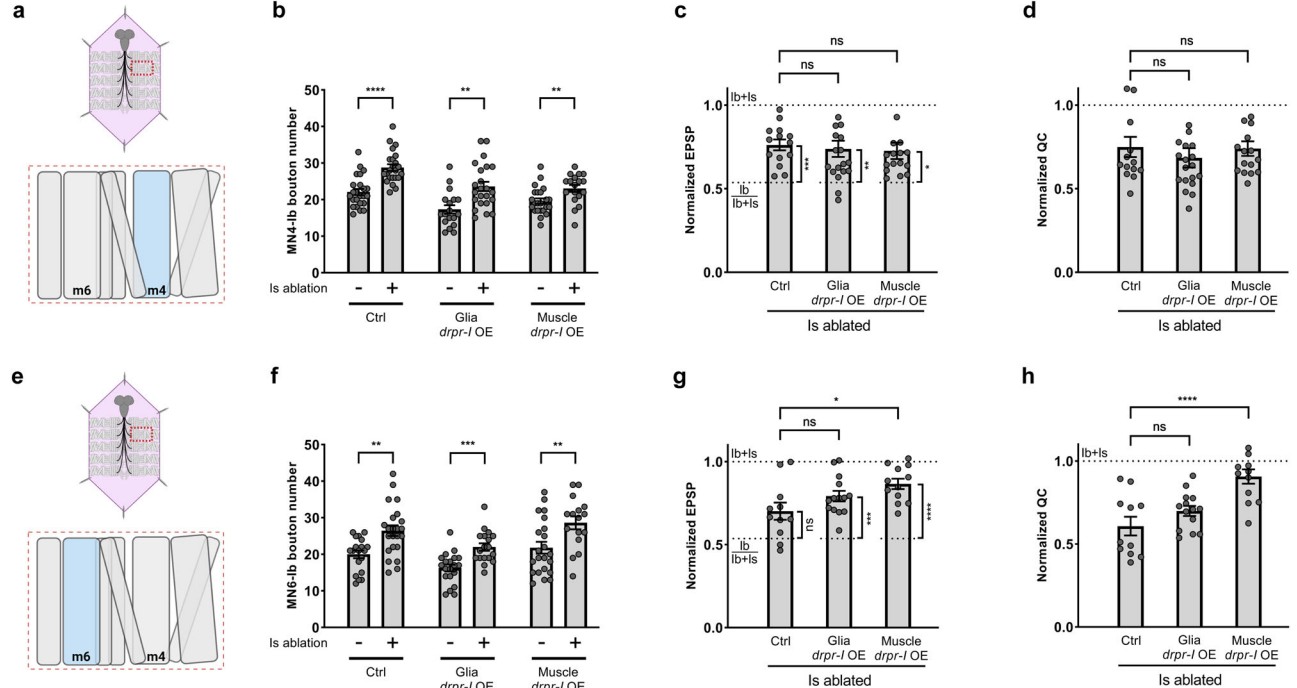

**Fig. 5 | Overexpression of Draper-I boosts cross-neuron plasticity of MN6-Ib.**
**a** Cartoon representation of a dissected larva (pink) and a hemisegment highlighted by dashed red rectangle. The target muscle examined in (**b–d**) is shown in blue. Cartoon is generated with Biorender. **b** Quantification of MN4-Ib bouton number in non-ablated and Is ablated larvae in control, glia *draper-I* overexpression, and muscle *draper-I* overexpression backgrounds. Control ($N = 24$ and 23 NMJs), t(45) = 5.321, $p < 0.0001$, unpaired $t$ test, two-tailed. Glia *draper-I* overexpression ($N = 18$ and 23 NMJs), t(39) = 3.557, $p = 0.001$, unpaired $t$ test, two-tailed. Muscle *draper-I* overexpression ($N = 23$ and 18 NMJs), t(39) = 2.844, $p = 0.0071$, unpaired $t$ test, two-tailed. **c** Quantification of normalized EPSP of MN4-Ib in Is ablated larvae in control, glia *draper-I* overexpression, and muscle *draper-I* overexpression backgrounds. F(2, 46) = 0.8117, $p = 0.4504$, One-way ANOVA. Is ablated control vs glia *draper-I* overexpression, $p = 0.9333$. Is ablated control vs muscle *draper-I* overexpression, $p = 0.8633$. Is ablated control vs Ib/Ib+Is, t(24) = 4.335, $p = 0.0002$, unpaired $t$ test, two-tailed. Is ablated in glia *draper-I* overexpression vs Ib/Ib+Is, t(28.3) = 3.001, $p = 0.0056$, unpaired $t$ test, two-tailed, with Welch's correction. Is ablated in muscle *draper-I* overexpression vs Ib/Ib+Is, t(26) = 2.547, $p = 0.0171$, unpaired $t$ test, two-tailed. **d** Quantification of normalized quantal content of MN4-Ib in Is ablated larvae in control, glia *draper-I* overexpression, and muscle *draper-I* overexpression backgrounds. F(2, 46) = 0.1383, $p = 0.8712$, One-way ANOVA. Is ablated control vs glia *draper-I* overexpression, $p = 0.6839$. Is ablated control vs muscle *draper-I* overexpression, $p = 0.9923$. For (**c**, **d**) N (NMJs) = 14, 19, 16.

**e** Cartoon representation of a dissected larva (pink) and a hemisegment highlighted by dashed red rectangle. The target muscle examined in (**f–h**) is shown in blue.. Cartoon is generated with Biorender. **f** Quantification of MN6-Ib bouton number in non-ablated and Is ablated larvae in control, glia *draper-I* overexpression, and muscle *draper-I* overexpression backgrounds. Control ($N = 19$ and 23 NMJs), t(40) = 3.493, $p = 0.0012$, unpaired $t$ test, two-tailed. Glia *draper-I* overexpression ($N = 20$ and 18 NMJs), t(36) = 4.103, $p = 0.0002$, unpaired $t$ test, two-tailed. Muscle *draper-I* overexpression ($N = 22$ and 16 NMJs), t(36) = 2.818, $p = 0.0078$, unpaired $t$ test, two-tailed. **g** Quantification of normalized EPSP of MN6-Ib in Is ablated larvae in control, glia *draper-I* overexpression, and muscle *draper-I* overexpression backgrounds. F(2, 34) = 4.361, $p = 0.0206$, One-way ANOVA. Is ablated control vs glia *draper-I* overexpression, $p = 0.2235$. Is ablated control vs muscle *draper-I* overexpression, $p = 0.0153$. Is ablated control vs Ib/Ib+Is, t(19) = 2.074, $p = 0.0519$, unpaired $t$ test, two-tailed. Is ablated in glia *draper-I* overexpression vs Ib/Ib+Is, t(22) = 4.041, $p = 0.0005$, unpaired $t$ test, two-tailed. Is ablated in muscle *draper-I* overexpression vs Ib/Ib+Is, t(20) = 5.061, $p < 0.0001$, unpaired $t$ test, two-tailed. **h** Quantification of normalized quantal content of MN6-Ib in Is ablated larvae in control, glia *draper-I* overexpression, and muscle *draper-I* overexpression backgrounds. F(2, 34) = 12.27, $p < 0.0001$, One-way ANOVA. Is ablated control vs glia *draper-I* overexpression, $p = 0.2951$. Is ablated control vs muscle *draper-I* overexpression, $p < 0.0001$. For (**g**, **h**), N (NMJs) = 11, 14, 12. Error bars indicate ± SEM, ns = non-significant, *$p < 0.05$, **$p < 0.01$, ***$p < 0.001$, ****$p < 0.0001$.

Finally, we overexpressed the less-studied *draper-III* in glial cells and muscles. MN4-Ib did not show a further increase of plasticity as we expected (Supplementary Fig. 10). For MN6-Ib, *draper-III* overexpression did not boost Ib structural compensation or EPSP amplitude, but muscle overexpression of *draper-III* caused a slight increase of quantal content (Supplementary Fig. 11), suggesting that Draper-III can trigger functional changes when overexpressed postsynaptically, specifically for MN6-Ib. Taken together, our data suggested, (1) Draper-I is the functional isoform in glial cells for cross-neuron plasticity; (2) muscles are capable of triggering cross-neuron plasticity; (3) increasing *draper-I* or *-III* can boost Ib plasticity selectively for MN6-Ib.

**Cross-neuron plasticity does not rely on Ib and Is co-innervation**
In the experiments above, only muscles that are co-innervated by both Ib and Is MNs were analyzed. Therefore, cell death and the axonal debris after Is MN ablation could be sensed by glial cells that wrap the axons as well as by postsynaptic muscles. We reasoned that if glial cells were the major player to transmit the signal, then co-innervation of Ib

and Is MNs on the same muscle should not be required to induce cross-neuron plasticity upon Is ablation (i.e. glial contact is sufficient to induce cross-neuron plasticity). To test this hypothesis, we examined MN11-Ib which innervates muscle 11 without Is co-innervation. We found a significant increase of bouton number, EPSP amplitude, and quantal content of MN11-Ib upon Is ablation (Fig. 7), suggesting that even without a co-innervating Is MN on the muscle target, Ib MNs still responded to Is ablation. These data are consistent with the model that glial cells play an important role in this process. In addition, this result further demonstrated that cross-neuron plasticity is a general mechanism in multiple MNs.

**Acute Is MN ablation induces functional plasticity**
Nervous system plasticity generally declines as animals age[34,40], but some plasticity mechanisms may extend until the later stages[41]. In our previous experiments, genetic ablation of Is MNs occurred in embryonic stages when the animals were still developing and synapses were undergoing extensive expansion and pruning[8]. Therefore, it is

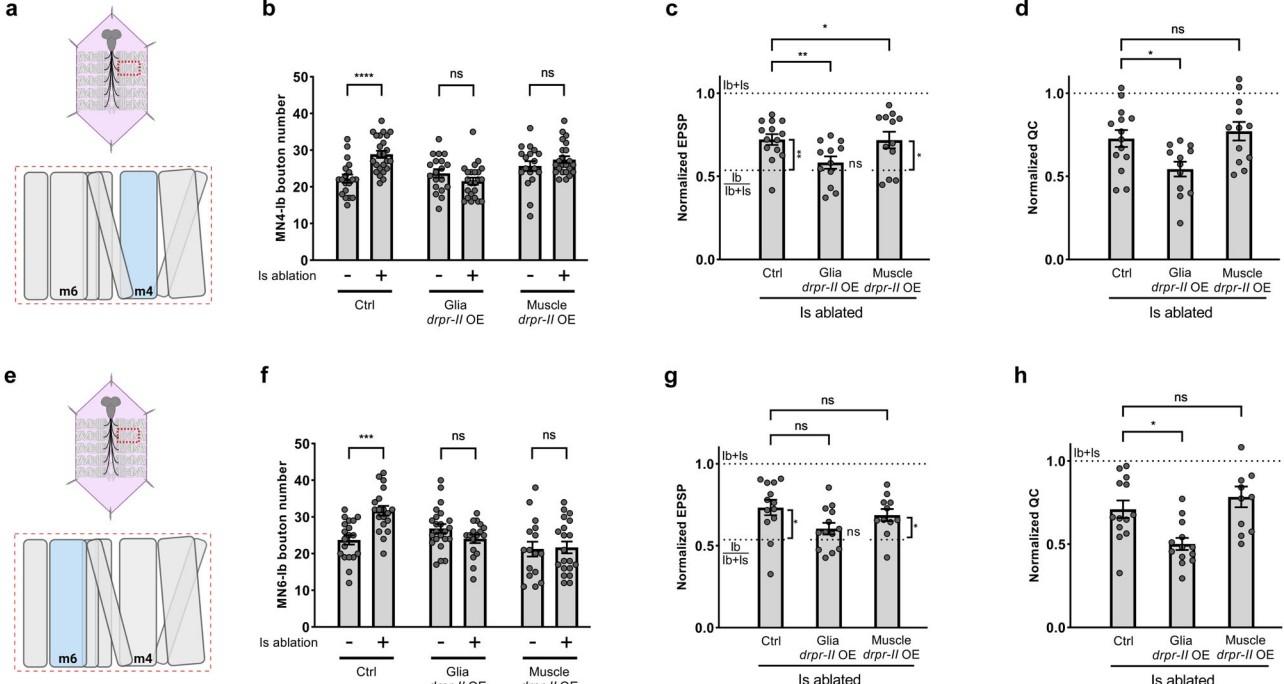

**Fig. 6 | Overexpression of Draper-II suppress cross-neuron plasticity. a** Cartoon representation of a dissected larva (pink) and a hemisegment highlighted by dashed red rectangle. The target muscle examined in (**b**–**d**) is shown in blue. Cartoon is generated with Biorender. **b** Quantification of MN4-Ib bouton number in non-ablated and Is ablated larvae in control, glia *draper-II* overexpression, and muscle *draper-II* overexpression backgrounds. Control ($N = 19$ and 24 NMJs), t(41) = 4.458, $p < 0.0001$, unpaired *t* test, two-tailed. Glia *draper-II* overexpression ($N = 19$ and 23 NMJs), t(40) = 1.440, $p = 0.1578$, unpaired *t* test, two-tailed. Muscle *draper-II* overexpression ($N = 18$ and 23 NMJs), t(39) = 1.114, $p = 0.2720$, unpaired *t* test, two-tailed. **c** Quantification of normalized EPSP of MN4-Ib in Is ablated larvae in control, glia *draper-II* overexpression, and muscle *draper-II* overexpression backgrounds. F(2, 35) = 1.535, $p = 0.2296$, One-way ANOVA. Is ablated control vs glia *draper-II* overexpression, $p = 0.0466$. Is ablated control vs muscle *draper-II* overexpression, $p = 0.9980$. Is ablated control vs Ib/Ib+Is, t(24) = 3.443, $p = 0.0021$, unpaired *t* test, two-tailed. Is ablated in glia *draper-II* overexpression vs Ib/Ib+Is, t(22) = 0.4045, $p = 0.6897$, unpaired *t* test, two-tailed. Is ablated in muscle *draper-II* overexpression vs Ib/Ib+Is, t(22) = 2.610, $p = 0.0160$, unpaired *t* test, two-tailed. **d** Quantification of normalized quantal content of MN4-Ib in Is ablated larvae in control, glia *draper-II* overexpression, and muscle *draper-II* overexpression backgrounds. F(2, 35) = 0.3700, $p = 0.6934$, One-way ANOVA. Is ablated control vs glia *draper-II* overexpression, $p = 0.0370$. Is ablated control vs muscle *draper-II* overexpression, $p = 0.8123$. For (**c**, **d**), N (NMJs) = 14, 12, 12. **e** Cartoon representation of a

dissected larva (pink) and a hemisegment highlighted by dashed red rectangle. The target muscle examined in (**f**–**h**) is shown in blue. Cartoon is generated with Biorender. **f** Quantification of MN6-Ib bouton number in non-ablated and Is ablated larvae in control, glia *draper-II* overexpression, and muscle *draper-II* overexpression backgrounds. Control ($N = 19$ and 19 NMJs), t(36) = 4.319, $p = 0.0001$, unpaired *t* test, two-tailed. Glia *draper-II* overexpression ($N = 23$ and 17 NMJs), t(38) = 1.626, $p = 0.1122$, unpaired *t* test, two-tailed. Muscle *draper-II* overexpression ($N = 16$ and 20 NMJs), t(34) = 0.1763, $p = 0.8611$, unpaired *t* test, two-tailed. **g** Quantification of normalized EPSP of MN6-Ib in Is ablated larvae in control, glia *draper-II* overexpression, and muscle *draper-II* overexpression backgrounds. F(2, 34) = 2.731, $p = 0.0795$, One-way ANOVA. Is ablated control vs glia *draper-II* overexpression, $p = 0.0680$. Is ablated control vs muscle *draper-II* overexpression, $p = 0.7116$. Is ablated control vs Ib/Ib+Is, t(21) = 2.606, $p = 0.0165$, unpaired *t* test, two-tailed. Is ablated in glia *draper-II* overexpression vs Ib/Ib+Is, t(21) = 1.008, $p = 0.3250$, unpaired *t* test, two-tailed. Is ablated in muscle *draper-II* overexpression vs Ib/Ib+Is, t(19) = 2.162, $p = 0.0436$, unpaired *t* test, two-tailed. **h** Quantification of normalized quantal content of MN6-Ib in Is ablated larvae in control, glia *draper-II* over-expression, and muscle *draper-II* overexpression backgrounds. F(2, 34) = 8.539, $p = 0.0010$, One-way ANOVA. Is ablated control vs glia *draper-II* overexpression, $p = 0.0130$. Is ablated control vs muscle *draper-II* overexpression, $p = 0.5614$. For (**g**, **h**), N (NMJs) = 13, 13, 11. Error bars indicate ± SEM, ns = non-significant, *$p < 0.05$, **$p < 0.01$.

important to understand whether cross-neuron plasticity is only inducible in early permissive stages, or if it is persists throughout development. To test if cross-neuron plasticity could be induced in later developmental stages, we established a heat-shock induced Is ablation system (Fig. 8a). Animals were raised at 18 °C and collected at different developmental stages from late-stage embryos to third instar larvae and subjected to heat-shock. Heat-shock induced expression of a Flippase transgene removes a stop codon flanked by two FRT sites, thus allowing the *Is-GAL4* to drive the expression of cell death genes. After Is ablation, animals were grown at 18 °C and examined at late third instar.

We first calculated the ablation efficiency and confirmed that our approach ablated approximately 80% of Is MNs, providing sufficient samples for analysis (Supplementary Fig. 12a). We then examined the NMJs and VNCs of animals with Is MNs ablated at different time points and confirmed that all debris from ablated Is MNs was removed by the time we assayed, which allowed us to faithfully study the Ib responses (Fig. 8b–g and Supplementary Fig. 12b–f). Examining muscle 4, we

found a significant increase of MN4-Ib bouton number only when Is MN ablation happened in embryonic stages, but not in larval stages (Fig. 8h). However, despite a lack of structural changes, MN4-Ib EPSP and quantal content were increased in all stages upon Is ablation (Fig. 8i, j and Supplementary Fig. 12g–i). These results suggested that (1) acute Is MN ablation induces functional plasticity of Ib MNs; (2) structural and functional plasticity may be regulated by different mechanisms; and (3) functional plasticity is not simply a consequence of more boutons.

## Cross-neuron plasticity enhances larval locomotion

In *Drosophila*, the Ib MNs are considered tonic neurons that provide sustained responses, while the Is MNs are phasic neurons that respond and adapt quickly[42]. It has been thought that the tonic Ib MNs are the major drive for normal larval behavior such as foraging and crawling, whereas the phasic Is MNs are responsible for quick actions such as escaping[35]. Here, we wondered what the behavioral consequences are upon Is ablation, focusing on behaviors that are attributed to both Ib

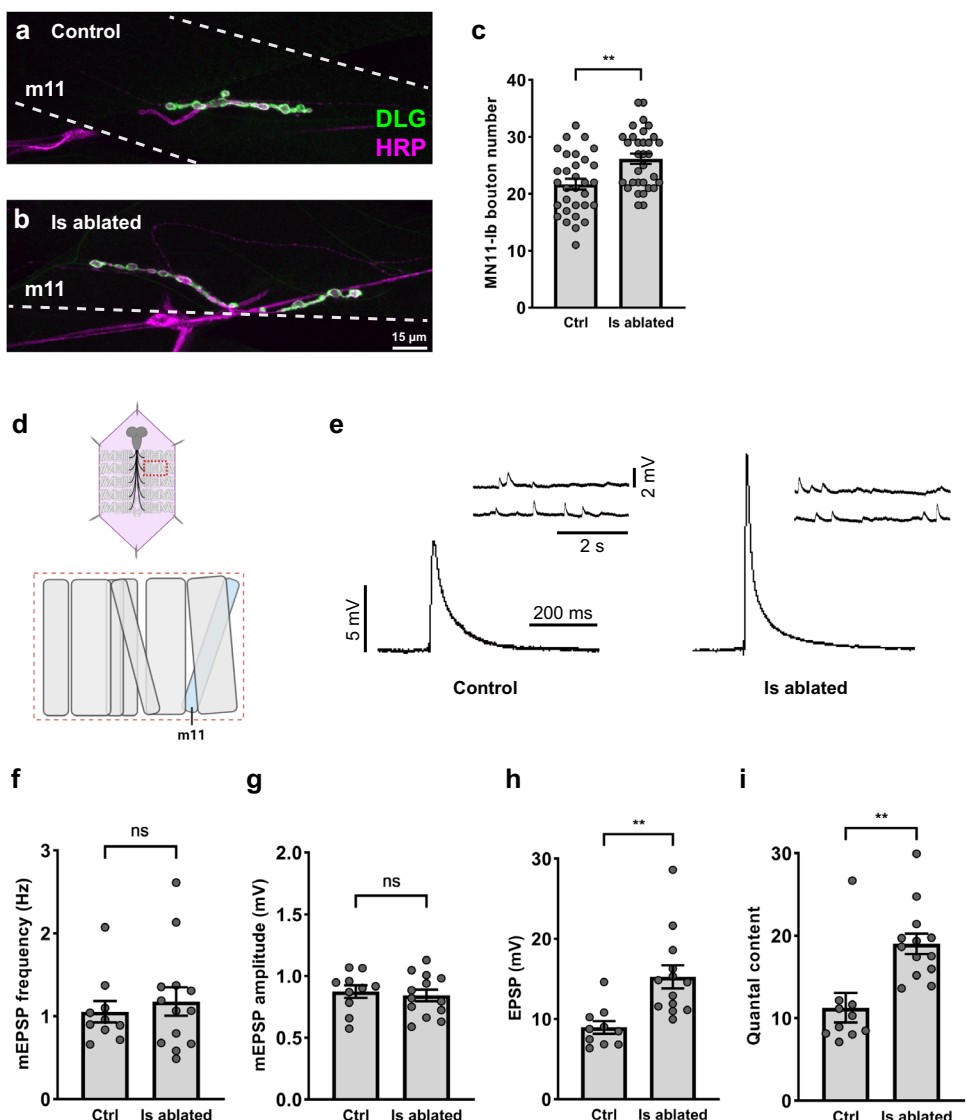

**Fig. 7 | MN11-Ib displayed cross-neuron plasticity upon Is ablation. a, b** NMJs of MN11-Ib in third instar control (*Is > GFP*) and Is ablated (*Is > GFP,hid,rpr*) larvae labeled with GFP (green) and HRP (magenta). Note the larger NMJs in Is ablated larvae. **c** Quantification of MN11-Ib bouton number in non-ablated (*N* = 31 NMJs) and Is ablated (*N* = 32 NMJs) larvae. t(61) = 3.408, *p* = 0.0012, unpaired *t* test, two-tailed. **d** Cartoon representation of a dissected larva (pink) and a hemisegment highlighted by dashed red rectangle. The target muscle examined in this figure is shown in blue (**f–i**). Cartoon is generated with Biorender. **e** EPSP and mEPSP recordings of muscle 11 from non-ablated and Is ablated larvae. **f** Quantification of mEPSP frequency. t(21) = 0.5408, *p* = 0.5943, unpaired *t* test, two-tailed. **g** Quantification of mEPSP amplitude. t(21) = 0.4446, *p* = 0.6611, unpaired *t* test, two-tailed. **h** Quantification of EPSP amplitude. t(18.02) = 3.840, *p* = 0.0012, unpaired *t* test, two-tailed, with Welch's correction. **i** Quantification of quantal content. t(21) = 3.657, *p* = 0.0015, unpaired *t* test, two-tailed. For (**f–i**), N (NMJs) = 10, 13. Error bars indicate ± SEM, ns = non-significant, **p < 0.01.

and Is activity. First, we analyzed the crawling behavior in freely moving larvae. Interestingly, we found that Is ablated animals had a faster crawling speed but turned less compared to wild type controls (Fig. 9a–c). We reasoned that the faster crawling speed was a consequence of Ib elevated activity upon Is ablation whereas the turning defect was due to the loss of multiple-targeted phasic Is MNs to aid synchronous movement to one side. To bolster this hypothesis, we examined *draper* mutant animals with or without Is ablation. We found that loss of *draper* blocked the increase of crawling speed upon Is ablation but did not affect the decreased turn frequency (Fig. 9a–c). Notably, *draper* mutant larvae showed a lower crawling speed, which might mask the speed increase upon Is ablation. Nevertheless, these data revealed cross-neuron plasticity at the behavioral level.

Next, we analyzed larval rolling behavior, a well-characterized escape behavior. When wild type larvae are confronted with noxious thermal stimulus, such as being immersed in a drop of water on a heating block, they exhibit a stereotyped rolling behavioral response[43]. We found that wild type larvae displayed sustained roll behavior, whereas Is ablated larvae showed extensive head stretch and bends but failed to perform complete rolling (Fig. 9d, e and Supplementary Movie 1). This result suggested that the Is MNs play an important role in the escape neural circuit, and Ib MN cross-neuron plasticity cannot compensate the behavioral changes, as expected due to their distinct circuit partners.

## Discussion

Neuronal cell death is a hallmark of aging, injury, and many neurodegenerative diseases and significant efforts have been made to delay, prevent, or ameliorate these incidents. Most neurons of the central nervous system cannot regenerate to restore function, but healthy bystander neurons may provide an alternative to overcome dysfunction after neuronal cell death. Indeed, several studies have reported

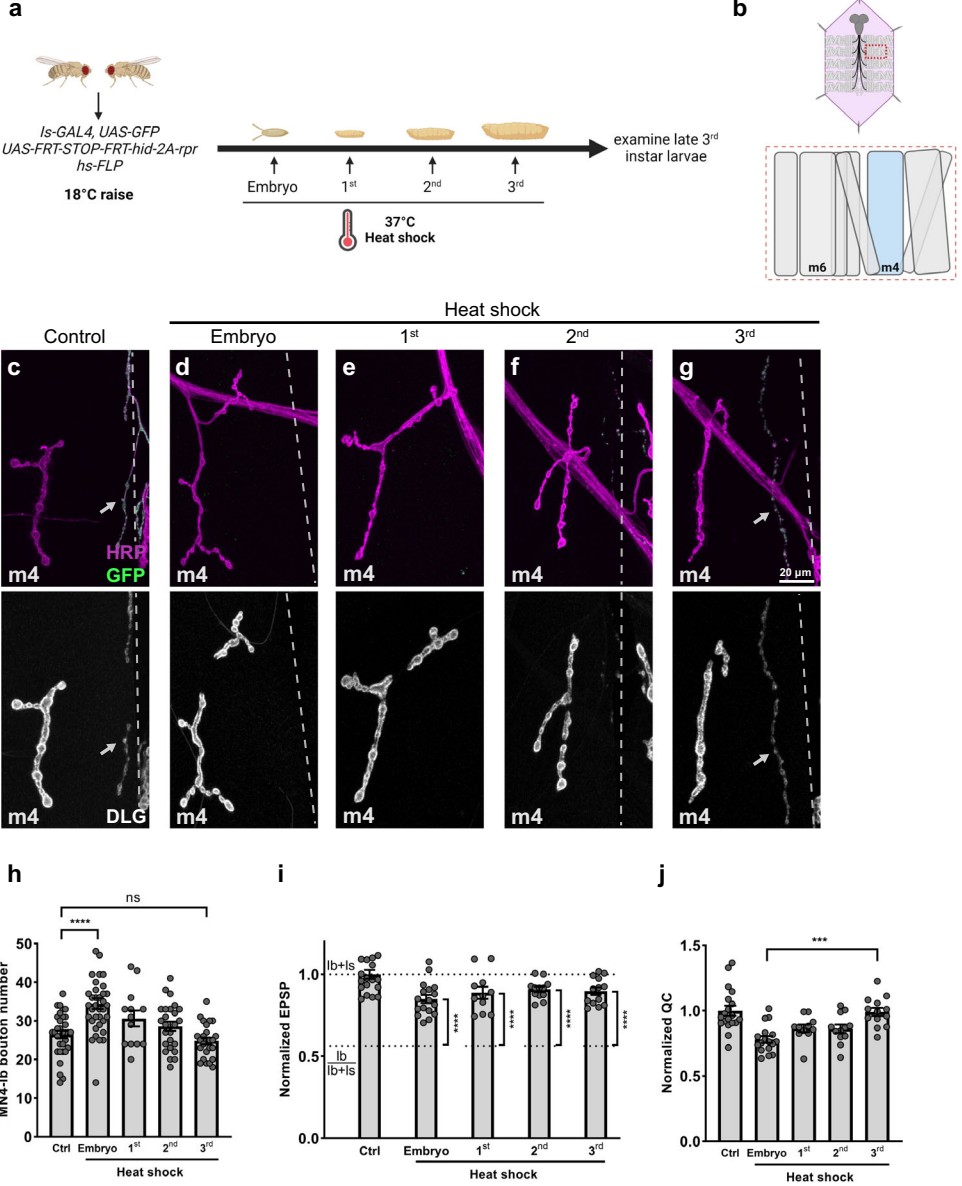

**Fig. 8 | Acute Is MN ablation induces functional plasticity, but not structural plasticity.** Schematic of heat-shock induced Is MN ablation protocol. **b** Cartoon representation of a dissected larva (pink) and a hemisegment highlighted by dashed red rectangle. The target muscle examined in this figure is shown in blue. Cartoon is generated with Biorender. NMJs of MN4-Ib in late third instar larvae (*hs-FLP,UAS-GFP/ + ;UAS-FRT stop FRT-hid-2A-rpr/+;Is-GAL4/+*) with (**c**) no heat-shock, (**d**) embryo heat-shock, (**e**) first instar heat-shock, (**f**) second instar heat-shock and (**g**) third instar heat-shock, stained with GFP (green), HRP (magenta), and DLG (gray). In embryos and first and second instar heat-shocked larvae, the Is NMJs were fully cleared, while Is synaptic debris remains in third instar heat-shocked larvae. Arrows indicate the Is MN (**c**) or Is MN debris (**g**). **h** Quantification of MN4-Ib bouton number in late third instar larvae with Is MNs ablated at different developmental stages. $F_{(4, 127)} = 10.23$, $p < 0.0001$, One-way ANOVA. Control vs embryo heat-shock, $p < 0.0001$. Control vs

first instar heat shock, $p = 0.3021$. Control vs second instar heat-shock, $p = 0.7310$. Control vs third instar heat-shock, $p = 0.8756$. N (NMJs) = 33, 36, 13, 25, 25. **i** Comparison of the normalized EPSP of late third instar larvae with Is MNs ablated at different developmental stages to Ib/Ib+Is baseline. Embryo heat-shock, $t_{(27)} = 7.126$, $p < 0.0001$, unpaired *t* test, two-tailed. First instar heat-shock, $t_{(21)} = 6.622$, $p < 0.0001$, unpaired *t* test, two-tailed. Second instar heat-shock, $t_{(22)} = 9.485$, $p < 0.0001$, unpaired *t* test, two-tailed. Third instar heat-shock, $t_{(23)} = 8.557$, $p < 0.0001$, unpaired *t* test, two-tailed. **j** Quantification of normalized quantal content of late third instar larvae with Is ablated at different developmental stages. $F_{(4, 67)} = 9.109$, $p < 0.0001$, One-way ANOVA. Embryo heat-shock vs third instar heat-shock, $p = 0.0002$. Non-significant for the others. This result suggested an increase of cross-neuron plasticity when acutely ablated Is MNs. For (**i, j**), N (NMJs) = 19, 17, 11, 12, 13. Error bars indicate ± SEM, ns = non-significant, \*\*\**p* < 0.001, \*\*\*\**p* < 0.0001.

that bystander neurons enhance their structural and functional properties when they detect loss of neighboring neurons[8,10]. These studies suggest a type of plasticity of the nervous system to maintain its functional state, which we termed as cross-neuron plasticity.

In this study, we found that the *Drosophila* engulfment receptor, Draper, and an interacting kinase, Shark, are required in glial cells for cross-neuron plasticity in Ib MNs after loss of Is MNs. Overexpression of *draper* boosted the compensatory changes in bystander Ib MNs, providing an exciting avenue to recover the functional defects. We also examined induction of cross-neuron plasticity at different time points and found that functional plasticity was inducible at all larval stages, suggesting cross-neuron plasticity does not simply reflect a highly plastic temporal window during embryonic development. Taken together, our study revealed important mechanistic insights into cross-neuron plasticity and established an entry point to understand how healthy bystander neurons respond to their dying neighbors.

One open question is what signal is sensed by glial cells or muscles to trigger cross-neuron plasticity. It is known that degenerating

neurons express or secrete "eat me" signals that are recognized by engulfment receptors on phagocytic cells, including glial cells[44,45]. In the *Drosophila* nervous system, the engulfment receptor, Draper, interacts with several "eat me" signals, including SIMU[46], pretaporter[47], and phosphatidylserine[48,49]. It will be of great interest to examine the potential ligands for Draper in cross-neuron plasticity.

Upon ligand binding, Draper is phosphorylated by Src42a and binds to Shark, which together become the signaling core for the clearance pathway[30]. Draper/Shark first activate Rac1 through DRK/DOS/SOS or dCed-12/MBC/Crk complexes to initiate glial membrane recruitment to engulf the debris from degenerating neurons[32,33]. In parallel, Draper/Shark activate the dJNK pathway and downstream dAP-1 and STAT92E transcription factors to drive the expression of engulfment genes[38,39]. This phagocytic pathway has been extensively studied during synaptic pruning and injury induced axon degeneration, and here we implicate Draper/Shark in removing neuronal debris during programmed cell death. As transcription factors of the vertebrate innate immune system, AP-1 and STATs regulate many essential cellular processes such as differentiation, proliferation, apoptosis, and expression of inflammatory cytokines and chemokines. Utilizing severed sensory neuron axons in adult fly wings, a recent study revealed that glial cells function through Draper->JNK->dAP-1 to suppress axon transport in bystander neurons, suggesting that glial cells detect and spread an injury signal to nearby healthy neurons[9]. Here, we explored this crosstalk in the neuromuscular system and found that cross-neuron plasticity in bystander neurons also relies on Draper signaling, which may function through a similar pathway discussed above. Loss of either *draper* or *shark* in glial cells completely suppressed structural and functional cross-neuron plasticity. Notably, our data also suggested a role of Draper signaling in muscles, as muscle *draper/shark* knockdown blocked compensation of the EPSP, and muscle overexpression of *draper-I* boosted plasticity of MN6-Ib. However, the muscle pathway might be less prominent than the glial pathway because structural plasticity or quantal content was not affected in muscle *draper/shark* knockdown. We reasoned that this could be due to the extensive contact between glial cells and the MNs along the nerve compared to the NMJ. In addition, the interaction between

different isoforms of Draper and how they collectively contribute to cross-neuron plasticity will need further examination.

Interestingly, our results suggested a positive role of the Draper/Shark pathway on bystander neurons during injury-induced responses. These positive effects seemingly contradict the Draper/Shark-mediated axon transport defects in bystander neurons when severing sensory neurons in the adult fly wing[9]. The opposing effects might be due to different mechanisms acting in sensory and motor systems, or more likely, to the different time length after ablation/injury. In the wing, axon transport defects were observed 3 h after injury, whereas we examined cross-neuron plasticity at least two days after ablation. Indeed, the behavioral defects caused by wing injury were fully recovered after 6 h, suggesting the bystanders might eventually compensate for the severed neurons.

Many exciting questions are spawned by our study including – what is the signal from glial cells to bystander neurons, and how do bystander neurons respond to such a signal to alter their morphology and functional output? Glial cells are intimately associated with neurons offering several opportunities for communication. First, glial cells and neurons may directly interact through cell surface receptors/ligands, such as Notch/Delta[50] and Nrx/Nlg[40]. Alternatively, glial cells may secrete molecules, like Wnt[51], Spz[45], and others, that bind to corresponding receptors on bystander neurons. It will be of great interest to examine if any of these signaling pathways are involved in cross-neuron plasticity. Notably, several lines of evidence suggest that bystander neurons may utilize distinct mechanisms for the structural and functional components of cross-neuron plasticity. In our previous study, MN12-Ib displayed robust structural plasticity but no functional plasticity when the Is neighbor was ablated, unlike MN6-Ib and MN4-Ib which showed structural and functional plasticity[8]. In the current study, when we acutely ablated Is MNs in third instar larvae, we only observed an increase of the EPSP, but no correlated structural bouton increase. Together these data suggest several non-exclusive models: (1) downstream of Draper/Shark, different mechanisms may be utilized to regulate structural or functional cross-neuron plasticity, (2) different MNs have differing capabilities to show either, or both, plasticity changes, and (3) structural plasticity may require time to add boutons

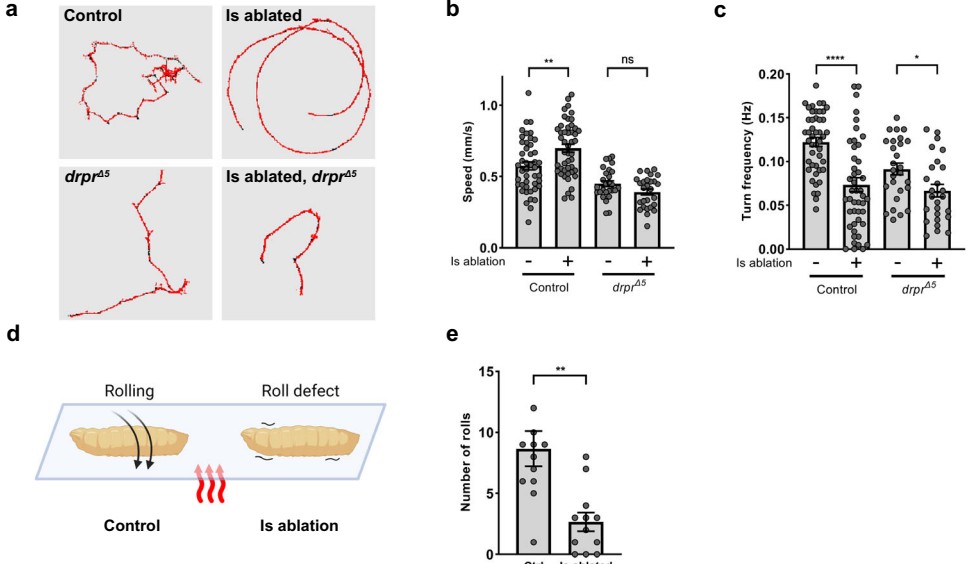

**Fig. 9 | Cross-neuron plasticity increases larval locomotion speed.**
**a** Representative crawling traces. **b** Quantification of crawling speed. Control, t(84) = 3.107, *p* = 0.0026, unpaired *t* test, two-tailed. *drpr*^Δ5^, t(52) = 1.991, *p* = 0.0517, unpaired *t* test, two-tailed. **c** Quantification of turn frequency. Control, t(72.67) = 4.518, *p* < 0.0001, unpaired *t* test, two-tailed, with Welch's correction. *drpr*^Δ5^, t(51) = 2.493, *p* = 0.0160, unpaired *t* test, two-tailed. For (**b**, **c**), N (larvae) = 45,

47, 27, 27. **d** Schematic of heat induced roll behavior difference in control and Is ablated larvae. Cartoon is generated with Biorender. **e** Quantification of the number of rolls of control and Is ablated larvae. t(16.59) = 3.647, *p* = 0.0021, unpaired *t* test, two-tailed, with Welch's correction. N (larvae) = 12, 12. Error bars indicate ± SEM, *\*p* < 0.05, *\*\*p* < 0.01, *\*\*\*\*p* < 0.0001.

gradually, whereas the functional plasticity may utilize established synaptic architecture to achieve a fast response. Indeed, active zones are highly dynamic and can acutely alter synaptic strength without the need of additional boutons. For example, the synaptic machinery, such as presynaptic active zone structural proteins[52,53] or postsynaptic neurotransmitter receptors[54], can increase their density to elevate synaptic release or the response to each synaptic vesicle, respectively. In addition, the properties of individual active zones, such as the size of the readily releasable pool and the release probability, may increase following perturbations[55–57]. Notably, in our previous study, the synaptic machineries in bystander neurons were not significantly altered, suggesting that functional cross-neuron plasticity may modify synaptic properties[8]. Examining these synaptic parameters will provide hints to explain the functional changes in bystander neurons.

Another important aspect to consider is to what extent plasticity mechanisms restore functionality. In many plasticity mechanisms such as PHP and PHD, MNs can fully restore their function to wild type levels[58]. In cross-neuron plasticity, Ib MNs did not fully restore the NMJ function to the wildtype "Ib+Is" level (Fig. 2j). However, in this paradigm, Is ablation happened in the late embryo when synaptic strength is significantly less compared to the third instar[59]. Therefore, Ib MNs could be tuned towards this lower "Ib+Is" level. The acute Is MN ablation experiment supported this hypothesis because there was a trend of increasing compensation when ablating Is MNs in later developmental stages (Fig. 8j).

Cross-neuron plasticity could be a broadly utilized mechanism and substantial evidence supports the potency of cross-neuron plasticity. For example, denervating muscles in both vertebrates and invertebrates lead to compensatory axon terminal expansion from healthy bystander neurons, which may eventually provide functional recovery[8,10,13,14,17]. On the behavioral level, severing sensory neurons in the adult fly wing causes immediate sensory defects that are rescued within 6 h[9]. In humans, sensory loss can lead to adaptation of brain circuits to utilize the remaining senses to navigate, a process known as cross-modal recruitment and compensatory plasticity (or cross-modal plasticity)[60,61]. Although the mechanisms underlying these phenotypes may vary, together they represent an intriguing, alternative paradigm that the nervous system can utilize to counteract synaptic dysfunction and neuronal death.

## Methods

### Fly and antibody reagents
The following fly stocks were used in this study: w[1118] [62], A8-GAL4 (Is-GAL4)[63], A8-LexA (this study), Repo-GAL4 and Mef2-GAL4 (gift from Kai Zinn), 10XUAS-mCD8::GFP (Bloomington Stock Center, BL #32184), UAS-hid,rpr[64], LexAOP-rpr[65], UAS-draper-RNAi (BL #67034), UAS-shark-RNAi (BL #42555), UAS-drpr-I (BL #67035), UAS-drpr-II (BL #67036), UAS-drpr-III (BL #67037), MHC-CD8::GCaMP6f-Sh[66], draper[AS] [67], hs-FLP (BL #28832), UAS-FRT stop FRT-hid-2A-rpr[68].

The following antibody reagents were used in this study: Mouse anti-Repo (Developmental Studies Hybridoma Bank, DSHB #8D12, 1:50), Chicken anti-GFP (gift from Michael Glotzer, 1:10,000), Rabbit anti-GFP (Thermo Fisher Scientific, #A11122, 1:500), Mouse anti-DLG (DSHB #4F3, 1:100), Rabbit anti-DLG[69] (1:40,000), Mouse anti-Draper (DSHB, #8A1-S, 1:30), Goat anti-HRP-Alexa Fluor 405 (Jackson ImmunoResearch, #123-475-021, 1:100), Goat anti-HRP-TRITC (Jackson ImmunoResearch, #123-025-021, 1:100), Goat anti-HRP-Alexa Fluor 647 (Jackson ImmunoResearch, #123-605-021, 1:100), Goat anti-mouse-Alexa Fluor 647 (Thermo Fisher Scientific, #A32728, 1:500), Goat anti-chicken-Alexa Fluor 488 (Thermo Fisher Scientific, #A11039, 1:500), Goat anti-rabbit-Alexa Fluor 568 (Thermo Fisher Scientific, #A11036, 1:500), Goat anti-rabbit-Alexa Fluor 488 (Thermo Fisher Scientific, #A11008, 1:500). Requests for resources and reagents should be addressed to Robert Carrillo (robertcarrillo@uchicago.edu).

### Cloning of A8-LexA
A 2 kb fragment from the A8-GAL4 promotor fragment was amplified using the following primers (aaaCCTAGGttatgtactccactattcttttttgc-taattttgcgc) and (aaaGGCCGGCCaagatatattaaaaaacatcaggaattatttctctc) and directionally cloned into the AvrII and FseI sites of the LexA::VP16 vector (gift from Claude Desplan). This construct was then integrated into the attP2 site on the third chromosome.

### Fly husbandry
Drosophila stocks and crosses were maintained at 25 °C except for the fly lines required for the heat-shock experiments. 6–8 females were mated with 3–5 males and transferred into new vials every day to ensure proper larval density. For heat-shock experiments, flies were mated and kept at 18 °C, and embryos or larvae at different developmental stages were collected for a 5 min 37 °C heat-shock in 1.5 mL Eppendorf tubes. Heat-shocked animals were transferred back to 18 °C until third instar. We included Is > GFP in all experiments to ensure that animals without Is ablation were co-innervated by both Ib and Is MNs on muscle 4, because approximately 20% of muscle 4 s lack Is innervation naturally[70]. Both genders were equally used in this study.

### Dissections, immunocytochemistry, and imaging
Dissections and immunostaining were performed as previously described[71]. Briefly, wandering third instar larvae were collected and dissected on sylgard plates in PBS. Samples were fixed by 4% paraformaldehyde for 20 min and then washed three times in PBT (PBS with 0.05% Triton X-100) for 15 min each. Samples were then blocked for 1 h in 5% goat serum (5% goat serum diluted in PBT) and incubated with primary antibodies at 4 °C overnight. The next day, primary antibodies were washed out with PBT and secondary antibodies were applied at room temperature for 2 h. Finally, samples were washed with PBT and mounted in Vectashield (Vector Laboratories). Images were acquired on a Zeiss LSM800 confocal microscope using either a 40X plan-neofluar 1.3 NA objective, or a 63X plan-apo 1.4 NA objective. The same imaging parameters were applied to samples from the same set of experiments. Images were then analyzed and processed in ImageJ.

### Electrophysiology
Third instar larvae were dissected in magnetic chambers using modified HL3 saline (70 mM NaCl, 5 mM KCl, 10 mM MgCl$_2$, 10 mM NaHCO$_3$, 5 mM trehalose, 115 mM sucrose, 5 mM HEPES) with 0.5 mM calcium. Segmental nerves were severed near the VNC and the brain and VNC were removed to prevent endogenous action potentials. Samples were examined under a Nikon FS microscope with a 40X long-working distance objective to locate the muscle fibers and axons. Muscles 4, 6 or 11 from abdominal segments A3 and A4 were chosen and impaled by a 10–30 MΩ sharp electrode filled with 3 M KCl. To elicit EPSPs from MN6-Ib, the entire segmental nerve was drawn into a suction electrode, whereas for MN4-Ib and MN11-Ib, the intersegmental nerve above muscle 5 was drawn. Each sample was first recorded for 1 min for miniature EPSPs (mEPSPs), followed by 2 min recordings of EPSPs at 0.2 Hz. 24 EPSPs were elicited and the larger 12 EPSPs were averaged to represent the mean EPSP. Due to the nonlinear summation of quantal content of large EPSPs, we corrected EPSP amplitude by the equations according to previous studies[72]. Quantal content was calculated by dividing the corrected EPSP amplitude by the mEPSP amplitude. However, in wild type animals, the EPSP and mEPSP are constituted by both Ib and Is MNs. Due to different synaptic vesicle sizes and glutamate receptor fields at each NMJ, this calculated quantal content is an estimate of the sum quantal content generated from both Ib and Is MNs. Electrophysiology signals were amplified by a MultiClamp 700B amplifier (Molecular Devices), digitized with a Digidata 1550B (Molecular Devices), and acquired in pCLAMP

10 software (Molecular Devices). Axon stimulation was delivered by a Master-9 stimulator (A.M.P.I.). Data was finally analyzed with Mini Analysis software (Synaptosoft).

## GCaMP imaging coupled with electrophysiology

Third instar *MHC-CD8::GCaMP6f-Sh* larvae were dissected and processed as described above. Larval fillets were visualized under a Nikon FS microscope with a 40X long-working distance objective and the GCaMP-positive Ib and Is NMJs were illuminated with an Aura II solid-state illuminator. Together with electrophysiology recordings, real time NMJ firing movies were recorded using a PCO Edge 4.2 camera and NIS-Elements Imaging Software (Nikon, version 5.00). Due to the different evoked thresholds of Ib and Is MNs, stimulating voltages were fine tuned to isolate Ib MN firing and Ib+Is firing[18,35]. EPSPs corresponding to specific MN firing combinations were categorized as Ib EPSP and Ib+Is EPSP. For each sample, Ib EPSP and Ib+Is EPSP were both recorded and the contribution of Ib MNs in wild type animals was calculated (Ib EPSP/Ib+Is EPSP).

## Behavioral assay

For rolling behavior, we followed the "Global Heat Plate Assay" protocol[43]. Briefly, a wandering third instar larva was placed in 200 µl of water at the center of a plastic 35 mm petri dish at room temperature and then the plate was transferred onto a 95 °C heat plate to elicit rolling behavior. Movies of the rolling behavior were captured by an RPi camera (Waveshare) and Point Grey FlyCap2 software. Only full lateral 360-degree rolls were counted by observing the dorsal trachea disappear under the larvae and emerge again on the opposite side. For the crawling assay, a third instar larva was placed on 2% agarose gel in a 100 mm petri dish at room temperature. Crawling trajectory was captured with a PiVR setup[73] at 50 FPS and the centroids of the larva were considered as larval positions. Sampling frames were selected every 50 frames and the total travel distance was the sum of the distance between every two larval positions in consecutive samples frames. Total travel distance was divided by time to calculate speed. A larva was considered to turn if the angle between two vectors formed by connecting three larval positions in consecutive sampling frames is between 45 degrees and 150 degrees, and turn frequency was calculated by dividing the number of turns by time. Note that the vector is at least 2 pixels in length to avoid discrepancies caused by body trembling.

## Statistical analyses

All statistical analyses were performed using Prism 8 and mean and SEM were reported. For each experiment, at least 10 samples were examined in at least two biological replicates. All data were assumed to follow a Gaussian distribution. When comparing the non-ablated versus Is ablated larvae under the same genetic background, two-tailed Student's *t* test was used (Welch's correction was used in case of unequal variance). When comparing across multiple conditions, One-way ANOVA with Turkey's test was performed.

## Reporting summary

Further information on research design is available in the Nature Portfolio Reporting Summary linked to this article.

## Data availability

Original data is provided in Source Data file. All other data is available upon request. Source data are provided with this paper.

## Code availability

The customized python code for data analyzation can be found at https://github.com/sihaohuanguc/larva_trajectory_process or Zenodo (https://doi.org/10.5281/zenodo.8011787).

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

## Acknowledgements

This work is supported by NSF IOS-2048080, NINDS R01 NS123439 01, and a UChicago Faculty Diversity Grant to R.A.C, F31NS120458 and T32 GM007183 to M.L-.R., and NIH R01NS070644 to Richard S. Mann (for support of L.V.). This work is also supported by funds from UChicago Biological Science Division, Committee of Developmental Biology and Department of Molecular Genetics & Cellular Biology. We thank the Bloomington Drosophila Stock Center (NIH P40OD018537) for fly lines. The monoclonal antibodies 4F3 and 8B12 were developed by Goodman, C., and the 8A1-S antibody was developed by Mary Logan, and they were obtained from the Developmental Studies Hybridoma Bank, created by the NICHD of the NIH and maintained at the University of Iowa, Department of Biology. We would like to thank Claude Desplan (New York University), Richard Mann (Columbia University), Robin E. Harris (Arizona State University) and Jean-Paul Vincent (the Francis Crick Institute) for sharing resources. We would also like to thank Richard Fehon, Ellie Heckscher, David Pincus, Audrina Daisy, Viola Nawrocka and members from the Carrillo laboratory for valuable discussions and comments. Cartoons in Figs. 2f, 3i, 4i, 5a, e, 6a, e, 7a, 8a, b, 9d, and Supplementary Figs. 4a, 5a, 6g, 7g, 8g, 9g, 10h, 11h, 12a, g were created with BioRender.com (license to Robert Carrillo).

## Author contributions

Y.W. and R.A.C designed research; Y.W., R. Z., P.T.T.V., L.V., and M-.L.R. performed experiments; Y.W., S.H. and J.A. analyzed data; Y.W. wrote the manuscript and J.A., M-.L.R. and R.A.C. edited the manuscript.

## Competing interests

The authors declare no competing interests.
