## [Peer Review File · Nature Communications]

Glial Draper signaling triggers cross-neuron plasticity in bystander neurons after neuronal cell death in *Drosophila*REVIEWER COMMENTS

Reviewer #1 (Remarks to the Author):

In this paper, Wang et al. describe how the process termed cross-neuron plasticity proceeds in after nearby neurons die. Interestingly, the authors find that in the absence of components like Draper or the Src family kinase member Shark, there is both an incomplete clearance of the dead and dying neurons. There are also consequences for the neuromuscular synapse developmentally and electrophysiologically.

The paper begins with the new observation that Draper is required for neuronal debris clearance in *Drosophila melanogaster* after genetic ablation of Is motor neurons. Is motor neuron ablation usually cause compensatory cross-neuron plasticity activity by Ib motor neurons, so the authors looked closely to see if this still occurs. It does not in draper KD loss of function, nor in shark KD loss of function. This suggests an inroad into how cross-neuron plasticity works on a molecular level.

The authors isolate the core effect of these genes to the glia. In follow-up experiments, the authors show that Draper-I overexpression is able to be instructive for inducing cross-neuron plasticity at that the signaling induced from the glia seems to be able to spread wherever the glia touch, including motor neurons on muscle 11, which are normally not innervated by Is.

The authors go on to examine the timing of this signaling process. They show that acute Is motor neuron ablation can induce this plasticity – but only the functional (electrophysiological) component, not the developmental/structural component. This result shows that the functional and structural components of cross neural plasticity are able to be uncoupled. Finally, the authors offer some evidence that cross neural plasticity is important for aspects of normal larval locomotion

This is a very nicely conducted study. It extends important concepts of structural and functional plasticity for tonic and phasic motor neurons that have not been delineated on molecular or cellular signaling levels before. This type of paper should be of wide interest to neuroscientists who are interested in both structural and developmental forms of plasticity. There are some issues regarding data and interpretation that should be addressed or tied up.

MAIN POINTS

1. Line 223-226 and Figure 3J, Figure S4, and Figure S6: it seems that collectively considering the data from all of these experiments, that knocking down Draper or Shark from muscles alone clearly has an effect on NMJ function. The argument about unchanged QC in lines 223-226 is a little difficult to evaluate without knowing the baseline minis in that experiment for the driver alone (the same baseline control is used for glia, muscle, and glia + muscle).

Clearly the glia data stand on their own. However, for this reviewer, it spurs a couple of questions. How do the authors suppose that the muscle promotes cross-neuron plasticity? (this issue is less addressed in the paper than the glial role). Second, why do the authors suppose that the muscle has an effect on its own, but the muscle plus glia effect does not seem to be additive beyond with the glia affect already is?

2. Lines 329 to 346 and Figure 7: If these ideas are correct about cross-neuron plasticity affecting

movement behavior then it should also be the case that this form of plasticity could be modified by the shark mutant or the Draper mutant. Are the authors able to rule this in or out? One alternative explanation would be that these movement phenotypes do not have anything to do with cross-neuron plasticity – rather, they may have to do with the promiscuous nature of Is bouton formation on a variety of muscles.

3. Data and statistical treatments – Figure 4B: In the control for this figure, it seems that cross-neuron plasticity is no longer taking place – if one looks at the individual experiment strictly from a statistical threshold point of view (however, the raw numbers for 4E and 4H do not seem all that different than 4B so the data might be on a statistical edge). Could it be the case that the numerical value is increased, but it does not achieve statistical significance? One suggestion would be for the authors to include exact p values for all figures where significance is not achieved, instead of n.s.

Separately, for Figure 4, what are the raw values for these overexpression conditions? It seems like we just have normalized values for everything in this figure. A table with actual values that readers can read and evaluate might be better.

4. Figure 5 (related to the above): This glial finding to MN 11 is impressive. This reviewer was wondering if the authors ever checked if the Draper-I overexpression in glia and if it similarly caused this same gain of neurotransmission (in MN 11) phenotype. It would be a great way to test if this Draper-mediated process from glia is instructive, potentially globally to all muscles contacted by glia.

MINOR POINTS

1. Figure 3: could one idea be that when Drapers is knocked down, that there is already a “maximal increase” in bouton number (compare glial condition to control in 3I)? By that interpretation, it really would not be a failure to implement cross-neuron plasticity program so much as a ceiling effect that occludes further increases by cross-neuron plasticity.

Figure S7J: This reviewer is a little puzzled because in this case, there does not seem to be any baseline compensation, including for the drpr-I overexpression condition.

Reviewer #2 (Remarks to the Author):

In Wang et al., the authors characterize the cellular mechanisms that contribute to "cross-neuron plasticity." Cross-neuron plasticity is the capacity of a surviving neuron to undergo structural and functional plasticity in response to the loss of their neighboring neurons. To characterize this process, the authors use the *Drosophila* larval NMJ and sophisticated genetics to ablate defined motoneuron types. They reasoned that glia might regulate cross-neuron plasticity because 1) they engulf cellular debris after injury and 2) are large enough to contact both the ablated and surviving cell type; thus, they might be transmitting signals to the surviving neuron to compensate for the loss of its neighbor. The authors show that loss of the engulfment receptor Draper from glia (and marginally from muscles) prevents cross-neuron structural and functional plasticity. Interestingly,

that structural plasticity is only induced during early development (embryo), whereas functional plasticity can be induced throughout larval life, suggesting that glia might transmit unique signals to induce these two forms of plasticity.

Overall, this story is very well done and the results are exciting. Cross-neuron plasticity has been observed in different model systems for 100 years, but this is the first story to demonstrate that the major cellular trigger for this plasticity comes from glia. As regrowth of CNS axons following injury is impaired due to accumulation of debris at the lesion site, these data point to the glial engulfment pathway as a major target to improve regenerative outcomes. I have several suggestions to clarify the data presented in this manuscript, which are very well done in general:

1. Figure 1A-E: It is strange to measure colocalization of neuronal debris with repo nuclei as an indication of failed engulfment. To truly see that, the authors should label the glial membranes with a complementary fluorophore (e.g. repo-lexA, lexAop-tdtomato) and show that the GFP is not internalized in the Draper mutants.
2. Figure 1H-I: With the images provided, it appears that dendritogenesis is severely impacted in the Draper mutants (panel H looks very different from panel F). The authors should do a statistical comparison of dendrite volume in controls versus mutants prior to ablation. Use of a myr::GFP (rather than plain GFP) may make the data more clear as well.
3. The authors use a series of cell-type specific Draper knockdowns (glia, muscle, and both) to assess where Draper is required for cross-neuron plasticity. As the authors have UAS-Draper lines, a nice complementary experiment would be to perform cell-type specific rescue in the Draper mutant backgrounds (DraperMUT, repo>UAS-Draper-I) to test if glial expression of Draper is sufficient to facilitate cross-neuron plasticity following ablation.

Minor Comments:

1. As the authors switch their model neurons a couple of times in the manuscript, a small cartoon in each figure highlighting which muscle and MN are being assessed would be helpful.
2. Typo: Discussion, line 352, should read "their structural and functional"

Overall, I commend the authors of their work. It's very well done, and merits publication in Nature Communications.

Reviewer #3 (Remarks to the Author):

Wang et al. investigate the signaling systems involved in what they refer to as "cross-neuron plasticity" using the *Drosophila* NMJ as a model system. Building off of previous work from their group and others, they find that genetic ablation of the Is motor neuron leads to both structural and functional changes in the remaining Ib motor neuron. Here, they test the hypothesis that Draper-mediated signaling is required for the cross-neuron plasticity observed, and find that indeed Drpr

signaling in glia (and likely muscle as well) is necessary. Beyond this major finding, they also show that *drpr* overexpression can “boost” this synaptic plasticity, that functional plasticity can be induced throughout larval development (while only early stages of *Is* ablation can induce structural plasticity), and show some changes in locomotor activity.

The manuscript is clear, well written, and the experiments are in general performed at a high quality. The major finding, that *Drpr* signaling is necessary for cross-neuron plasticity, is of significant importance and impact to the field, as it reveals the first insights into how plasticity induced by ablation of one motor neuron is propagated to neighboring neurons. This result is not unexpected, given the seminal work by Marc Freeman showing that glia and *Drpr* signaling similarly propagate adaptations in bystander neurons in the wing after injury, but it is an important finding nonetheless. It remains unclear how *Drpr* operating in glia/muscle signals to neurons, and mechanistic insight into this would really add a lot to this manuscript. There are also some significant weaknesses in the areas discussed below that need to be addressed, particularly in the authors approach and analysis of synaptic function. Nonetheless, establishing *drpr*-mediated signaling and role of glia and muscles in transmitting cross-neuronal plasticity generates an exciting foundation to understand how healthy neurons adapt to neuronal death and will be of broad interest and impact to the field.

Major points:

1. Electrophysiological analysis: There are several significant issues with the electrophysiology presented in this manuscript.

1A: The authors use an indirect measure of *Ib/Ib+Is* to attempt to estimate EPSP amplitude coming from *Ib* and *Is*. It seems that throughout this manuscript, the authors should just report *Ib* EPSP amplitude, which is all they are really interested in. What is the point of showing the ratio between *Ib/Ib+Is*? The functional plasticity the authors are interested in is just *Ib* at baseline and after *Is* ablation in all of the experiments presented. Hence, it would be clearest and most straightforward to just show *Ib* EPSP amplitudes and ignore the *Ib+Is* blended EPSP at baseline. For example, in Fig. 2F, only *Ib* EPSPs should be shown and made clear based on their simultaneous optical recordings.

This is assuming the authors can be absolutely sure that they are measuring just the *Ib* EPSP with their threshold manipulations. Ideally the key experiments would be done as in Fig. S1 visualizing the Ca response from *Ib* only to ensure they are really just measuring *Ib* EPSPs. Even better would be to use one of the more recent approaches the field has employed to selectively stimulate *Is* or *Ib* (optogenetics, botulinum toxin, etc).

1B: Quantal content cannot be accurately compared between *Is* intact vs *Is* ablated. As the authors know, mEPSP amplitude is dramatically different (over 50%) between the two inputs due to differences in synaptic vesicle size and the glutamate receptor fields at each NMJ. Hence, even if the authors could accurately measure *Ib* EPSP amplitude (which it is not clear as discussed above), quantal content cannot be accurately determined by simply dividing the EPSP amplitude from the averaged mEPSP amplitude (which is a weighted average of minis coming from *Is* and *Ib*). While the quantal content value is not absolutely critical for the major findings in this manuscript, it is suggested that the authors explain this limitation in their approach and refer to their quantal content value as “apparent quantal content” or “estimated” quantal content.

1C: It is strange that in Fig. S2, there is no change in mini amplitude or frequency with or without *Is* ablation. Do the authors have an explanation for this? From the Han et al. 2022 paper and others, it

is very clear that minis are much larger at Is NMJs, so in Is ablated conditions mEPSP amplitude should be expected to be significantly smaller. Similarly, there should be fewer minis released in the absence of Is, so mEPSP frequency should also be reduced. There are also differences in the Ib EPSP amplitude compared to the Is contribution presented in this manuscript compared to previous studies. All of this was worked out in the Newman et al. 2017 and Han et al. 2022 studies. Do the authors think there is some functional plasticity induced in mini freq/amp that were not observed in the previous studies?

2. Interpretation of Drpr overexpression results: The authors appear to really stress that Drpr is needed in glial cells to signal Is ablation and promote cross-neuron plasticity. But in most of their results, Drpr also appears to be needed in muscle as well. It is not clear why they emphasize Drpr being needed in glia so prominently.

In addition, what does “enhanced plasticity” from Drpr overexpression actually mean? If the system wants to restore NMJ function to baseline (Is+Ib), shouldn’t EPSP values from Ib be potentiated much more than what they observe? Even with drpr overexpression, EPSP amplitude from Ib is not sufficient to recover to baseline states (Is+Ib). So the reader is left wondering how to interpret these results; what signal is actually being sensed, and why would Drpr levels be limiting in this pathway? What parameter is actually being targeted in this “cross neuron plasticity”?

Most importantly, the authors move to the muscle 6 NMJ to do this experiment, and the rationale for this is not established. They mention their apparent concern that the muscle 4 NMJ may be “significantly higher...and may not show further enhancement” line 262. This possibility does not relieve them from doing and presenting the experiment at muscle 4 and, if they do not see any change in plasticity, they might then go to muscle 6. Indeed, it’s not clear why they just ignored the muscle 4 NMJ that they did all of their previous experiments in. At minimum, all of the experiments shown in Fig. 4 should be first done and presented at the muscle 4 NMJ, and then they can also show the muscle 6 data. If Drpr overexpression has no impact on muscle 4 plasticity, that is an important result that needs to be shown up front.

3. As the authors note there are two Is motor neurons that innervate separate sets of muscles. It would be of interest to manipulate both Is motor neurons and more deeply probe plasticity. In particular, the following experiments should be considered:

- At muscles innervated by the non-ablated Is input, does the Ib (or Is) motor neurons adapt or change when Is is ablated at the other muscle segments?
- What happens when both Is inputs are ablated? Does the Ib motor neuron show “enhanced cross neuron plasticity”, both in terms of structural and functional plasticity?
- How is larval locomotor behavior altered when both Is motor neurons are ablated (experiments like the ones presented in Fig. 7)?

4. Behavioral experiments presented in Fig. 7 – an absolutely crucial experiment to properly interpret the behavioral results is to perform the same experiments in baseline and Is ablated animals in a Drpr mutant background to determine whether the NMJ plasticity is actually responsible for the behavioral changes (as opposed to non-specific developmental or pre-motor circuit changes).

Minor points:

1. Some changes to the text need to be made:

Abstract line 40: "...axon terminal size and activity" ..- This should be "axon terminal NUMBER and activity" if the authors are referring to the moderate increase in bouton number.

Line 66: Should also cite PMID: 29748610

Typos: Line 67 "the" should be "for"; line 79 remove "the";

2. In some areas of the manuscript, the authors use the term "homeostatic", where it seems the term "adaptive" is more appropriate. Homeostatic refers to restoring fully baseline states, while the functional changes shown appear to enhance release, synaptic transmission is not restored to wild type baseline states (normal EPSP amplitudes in Is+Ib).

REVIEWER COMMENTS

Reviewer #1 (Remarks to the Author):

In this paper, Wang et al. describe how the process termed cross-neuron plasticity proceeds in after nearby neurons die. Interestingly, the authors find that in the absence of components like Draper or the Src family kinase member Shark, there is both an incomplete clearance of the dead and dying neurons. There are also consequences for the neuromuscular synapse developmentally and electrophysiologically.

The paper begins with the new observation that Draper is required for neuronal debris clearance in *Drosophila melanogaster* after genetic ablation of Is motor neurons. Is motor neuron ablation usually cause compensatory cross-neuron plasticity activity by Ib motor neurons, so the authors looked closely to see if this still occurs. It does not in draper KD loss of function, nor in shark KD loss of function. This suggests an inroad into how cross-neuron plasticity works on a molecular level.

The authors isolate the core effect of these genes to the glia. In follow-up experiments, the authors show that Draper-I overexpression is able to be instructive for inducing cross-neuron plasticity at that the signaling induced from the glia seems to be able to spread wherever the glia touch, including motor neurons on muscle 11, which are normally not innervated by Is.

The authors go on to examine the timing of this signaling process. They show that acute Is motor neuron ablation can induce this plasticity – but only the functional (electrophysiological) component, not the developmental/structural component. This result shows that the functional and structural components of cross neural plasticity are able to be uncoupled. Finally, the authors offer some evidence that cross neural plasticity is important for aspects of normal larval locomotion

This is a very nicely conducted study. It extends important concepts of structural and functional plasticity for tonic and phasic motor neurons that have not been delineated on molecular or cellular signaling levels before. This type of paper should be of wide interest to neuroscientists who are interested in both structural and developmental forms of plasticity. There are some issues regarding data and interpretation that should be addressed or tied up.

MAIN POINTS

1. Line 223-226 and Figure 3J, Figure S4, and Figure S6: it seems that collectively considering the data from all of these experiments, that knocking down Draper or Shark from muscles alone clearly has an effect on NMJ function. The argument about unchanged QC in lines 223-226 is a little difficult to evaluate without knowing the baseline minis in that experiment for the driver alone (the same baseline control is used for glia, muscle, and glia + muscle).

Reply: (Figure 3J is current 3k, S6 is current S5) To avoid the effects caused by introducing transgenes (*GAL4* or *UAS*), EPSPs and QCs from Is ablated animals are first normalized to the non-ablated controls with the same genetic manipulation. For example, “muscle *drpr* knock down with Is ablation” is normalized to “muscle *drpr* knock down without Is ablation”. These two conditions both have *mef2-GAL4*, *UAS-draper-RNAi*, *Is-LexA*, except that the Is ablated condition also carries *LexAOP-rpr* to ablate the Is MNs. Thus, this normalization should eliminate the effects caused by these transgenes, and the effect caused by overexpression itself. Then, the normalized EPSPs are compared across different genetic manipulations, as

well as a common baseline Ib/Ib+Is control (for example, Figure 3k). However, the normalized QCs are only compared across different genetic manipulations, but not compared to a common baseline control (for example, Figure 3l). This comparison between normalized QC is our bona fide indicator to evaluate if *draper/shark* is required for cross-neuron plasticity.

However, we do agree with the Reviewer that normalizing to a non-ablated control is not the best estimate of Ib activity. Knowing the baseline Ib EPSP and mEPSP in each genetic manipulation will be ideal. To achieve this, we recently used Botulinum neurotoxin (*UAS-BoNT-C*, gift from Dr. Dion Dickman (Han et al., 2022)) to block both spontaneous and evoked neurotransmitter release. Unfortunately, driving BoNT-C using our *Is-GAL4* causes neuronal cell death or severe undergrowth (Figure below). Thus, we cannot ensure that the activity from this genotype reflects a “wild type Ib”, because the Ib MNs may already compensate for the Is changes.

Is>GFP, BoNT-C

Clearly the glia data stand on their own. However, for this reviewer, it spurs a couple of questions. How do the authors suppose that the muscle promotes cross-neuron plasticity? (this issue is less addressed in the paper than the glial role).

Reply: We thank the Reviewer for mentioning the potential of muscles to promote cross-neuron plasticity. We think the muscle is not required, but may be sufficient to trigger plasticity. Because 1) knocking down *draper/shark* in muscles does not affect the increased Ib bouton number or QC (suggesting it is not required); 2) overexpressing *draper-I* in muscle boosts compensation of MN6-Ib (suggesting it may be sufficient). The reason could be due to the extensive contact between glial cells and MNs along the nerve compared to the NMJ. We highlight this hypothesis in Discussion section (Line 398-406).

Second, why do the authors suppose that the muscle has an effect on its own, but the muscle plus glia effect does not seem to be additive beyond with the glia affect already is?

Reply: We appreciate the Reviewer for bringing up this interesting point. We assume that the reviewer is referring to the reduced compensation of normalized EPSP in current Figure 3k and 4k. Here, single *draper/shark* knockdown in glial or muscle does not appear to be additive compared to the double knockdown. This could be due to the level of “baseline”. The Ib/Ib+Is baseline in the “normalized EPSP” graphs is generated in a GCaMP background. This baseline is a good reference to determine significant plasticity changes of the EPSP, but may not be

ideal to quantitatively calculate the level of compensation, as it could be slightly offset from the different genetic backgrounds. In addition, we think the glial effect is much stronger by itself and could mask the muscle effect in the double knockdown. Indeed, in double knockdown, the compensation of Ib bouton number and the compensation of QC show the same level as glial single knockdown. Thus, further additive effects would be hard to visualize.

2. Lines 329 to 346 and Figure 7: If these ideas are correct about cross-neuron plasticity affecting movement behavior then it should also be the case that this form of plasticity could be modified by the shark mutant or the Draper mutant. Are the authors able to rule this in or out? One alternative explanation would be that these movement phenotypes do not have anything to do with cross-neuron plasticity – rather, they may have to do with the promiscuous nature of Is bouton formation on a variety of muscles.

Reply: (Figure 7 is current Figure 9) We thank the Reviewer for this suggestion, and we performed behavior analyses on *draper* mutant animals with or without Is ablation (Line 338-343). We found that the increased crawling speed in Is ablated animals is no longer observed in *draper* mutants, but the turn frequency is not affected. However, we found that the *draper* mutant larvae crawl in an unsynchronized manner with a slower speed, which may mask the compensation from cross-neuron plasticity. Note that we also optimized our customized code to be able to filter out the unsynchronized body contraction (which sometimes could be tracked as a “turn”). We re-processed the data from control and Is ablated animals using the updated code.

In addition, we are intrigued by the Reviewer’s idea that the increased speed upon Is ablation could be due to losing the promiscuous innervation of Is MNs. Indeed, the promiscuous innervation pattern of Is MN could reduce moving speed as it could generate unsynchronous movement during locomotion. The loss of speed compensation in *draper* mutant larvae may help distinguishing these two hypotheses but it is of great interest to further explore this question.

3. Data and statistical treatments – Figure 4B: In the control for this figure, it seems that cross-neuron plasticity is no longer taking place – if one looks at the individual experiment strictly from a statistical threshold point of view (however, the raw numbers for 4E and 4H do not seem all that different than 4B so the data might be on a statistical edge). Could it be the case that the numerical value is increased, but it does not achieve statistical significance? One suggestion would be for the authors to include exact p values for all figures where significance is not achieved, instead of n.s.

Reply: (Due to re-organization of graphs, we added Figure 4B, E, H below from the previous manuscript draft for easier comparison)

We thank the Reviewer for pointing this out. Indeed, the p value here is 0.0519 and it is on the statistical edge. This is due to the fact that MN6-Ib does not compensate very well (also the reason we chose it in this overexpression experiment). In the current manuscript, we report exact p value, t value and degree of freedom in figure legends.

Separately, for Figure 4, what are the raw values for these overexpression conditions? It seems like we just have normalized values for everything in this figure. A table with actual values that readers can read and evaluate might be better.

Reply: We appreciate the Reviewer's suggestion. Original electrophysiology values are now reported as supplementary data.

4. Figure 5 (related to the above): This glial finding to MN 11 is impressive. This reviewer was wondering if the authors ever checked if the Draper-I overexpression in glia and if it similarly caused this same gain of neurotransmission (in MN 11) phenotype. It would be a great way to test if this Draper-mediated process from glia is instructive, potentially globally to all muscles contacted by glia.

Reply: We thank the Reviewer's advice. The original reason we examined MN6-Ib but not MN4-Ib or MN11-Ib was to avoid a ceiling effect, because MN4-Ib and MN11-Ib already shows strong compensation even without *draper* overexpression. Per the request of the third Reviewer, we overexpressed three *draper* isoforms and examined the responses from MN4-Ib (Line 245-286). The results confirmed our hypothesis – overexpression of *draper-I* does not trigger a further increase of compensation of MN4-Ib. We would hypothesize that MN11-Ib, similar to MN4-Ib, would not increase compensation because it already shows robust plasticity. In addition, recording from muscle 11 is also not trivial due to the difficulty of accessing the muscle, and once impalement is achieved, keeping a good seal is not easy. Examining MN11-Ib in overexpression conditions (control, glial overexpression, muscle overexpression with and without Is ablation) of all three isoforms would significantly delay our revision timeline. But in future studies, this would be a great way to examine the scale of Draper-mediated cross-neuron plasticity.

MINOR POINS

1. Figure 3: could one idea be that when Drapers is knocked down, that there is already a “maximal increase” in bouton number (compare glial condition to control in 3I)? By that

interpretation, it really would not be a failure to implement cross-neuron plasticity program so much as a ceiling effect that occludes further increases by cross-neuron plasticity.

Reply: We agree with this possibility that the increase of bouton number in glial conditions could mask the further increase upon Is ablation. We mentioned this possibility in Line 211-213.

Figure S7J: This reviewer is a little puzzled because in this case, there does not seem to be any baseline compensation, including for the *drpr-I* overexpression condition.

Reply: (Figure S7J is current Figure S9I) We do agree with the Reviewer that in some cases, increasing or decreasing bouton number will not impact EPSP due to some homeostatic mechanisms (Goel et al., 2019). In this case, we do not know how *Draper* affects neurotransmission and if *draper* mutation blocks homeostatic scaling as well. However, as we are comparing the normalized EPSP and QC, the increased baseline EPSP in overexpression conditions should not affect our conclusions.

Reviewer #2 (Remarks to the Author):

In Wang et al., the authors characterize the cellular mechanisms that contribute to "cross-neuron plasticity." Cross-neuron plasticity is the capacity of a surviving neuron to undergo structural and functional plasticity in response to the loss of their neighboring neurons. To characterize this process, the authors use the *Drosophila* larval NMJ and sophisticated genetics to ablate defined motoneuron types. They reasoned that glia might regulate cross-neuron plasticity because 1) they engulf cellular debris after injury and 2) are large enough to contact both the ablated and surviving cell type; thus, they might be transmitting signals to the surviving neuron to compensate for the loss of its neighbor. The authors show that loss of the engulfment receptor *Draper* from glia (and marginally from muscles) prevents cross-neuron structural and functional plasticity. Interestingly, that structural plasticity is only induced during early development (embryo), whereas functional plasticity can be induced throughout larval life, suggesting that glia might transmit unique signals to induce these two forms of plasticity.

Overall, this story is very well done and the results are exciting. Cross-neuron plasticity has been observed in different model systems for 100 years, but this is the first story to demonstrate that the major cellular trigger for this plasticity comes from glia. As regrowth of CNS axons following injury is impaired due to accumulation of debris at the lesion site, these data point to the glial engulfment pathway as a major target to improve regenerative outcomes. I have several suggestions to clarify the data presented in this manuscript, which are very well done in general:

1. Figure 1A-E: It is strange to measure colocalization of neuronal debris with repo nuclei as an indication of failed engulfment. To truly see that, the authors should label the glial membranes with a complementary fluorophore (e.g. *repo-lexA*, *lexAop-tdtomato*) and show that the GFP is not internalized in the *Draper* mutants.

Reply: We appreciate the Reviewer's great observation and suggestion. The reason that we show co-localization with Repo from the cross section is to highlight that the GFP signal is not from an axon. And indeed, the co-localization between GFP and Repo suggests that these debris are engulfed, but not cleared or digested (we claimed in the manuscript that *draper* mutant "failed the clearance" but not "failed the engulfment").

In the larval peripheral nervous system, there are three types of glial cells surrounding these MN axons and forming tight connections. Wrapping glia is the innermost layer that directly contacts axons, then there are perineurial glia and sub-perineurial glia forming the outer layers (Figure A). Because the wrapping glia have direct contact with both Ib and Is MNs, we hypothesize that they are responsible for engulfment and signal transduction in a Draper dependent manner. In *draper* mutants, wrapping glia cannot efficiently engulf these Is MN debris, and thus the debris diffuse into outer layers and are picked up by other glial cell types. Several lines of evidence support this hypothesis. First, from the cross section in Figure 1, the GFP signal accumulates in the outer layer of the segmental nerve, where the sub-perineurial glia and perineurial glia are located, suggesting that the debris is engulfed. If the engulfment step had failed in all glial cell types, we would expect a more diffused GFP signal. In addition, we found that this GFP positive glial cell type sometimes extend their processes into the NMJ area (Figure B). According to the literature, these processes are features of the sub-perineurial glia but not perineurial glia (Fuentes-Medel et al., 2009; Kerr et al., 2014). Therefore, we think the sub-perineurial glia are competent to engulf debris in a non-Draper dependent manner and may serve as a backup for clearing debris. However, sub-perineurial glia do not appear able to digest the debris and transmit the injury signal to induce cross-neuron plasticity.

Panel a has been reacted due to 3rd Party Rights

In our manuscript, we chose not to discuss this hypothesis because we lack direct evidence. We contemplated addressing this question by dual labeling the Is MN and specific glia subtype. However, the genetics would require exhaustive recombination since this line requires: (1) glia subtype specific-GAL4, UAS-RFP, Is-LexA, LexAOP-GFP (for dual labeling); (2) LexAOP-rpr (for Is ablation); and (3) homozygous *draper* mutant background. An alternative approach is to examine cross-neuron plasticity in glia cell type-specific *draper* knockdown. If the hypothesis is correct, removing *draper/shark* from wrapping glia will block plasticity but removing *draper/shark* from other glial cell types (including glial cell types in VNC) will have no effect. However, these analyses would require extensive electrophysiology recordings, and the main thrust of this study was not to identify the specific glial cell type that underlies cross-neuron plasticity. Thus, we think it is beyond the scope of this paper.

2. Figure 1H-I: With the images provided, it appears that dendritogenesis is severely impacted in the Draper mutants (panel H looks very different from panel F). The authors should do a statistical comparison of dendrite volume in controls versus mutants prior to ablation. Use of a *myr::GFP* (rather than plain GFP) may make the data more clear as well.

Reply: We appreciate the careful examination of our data. We did not show the most representative images/panels when preparing these figures because we tried to avoid the ventral Is MN cell bodies and projections. This forced us to include less confocal slices when

performing Z projections for this specific sample. We have updated the representative image in the current manuscript to better reflect the data.

However, although the dendrites look similar from most of our Z projections, the question brought up by this Reviewer is very interesting – whether loss of *draper* affects dendrite structure/volume in larvae. Previous studies did not examine the role of Draper in larval central nervous system development. Below, we estimated the dendritic volume of Is MNs in hemisegment A1, using our previously acquired imaging samples. We created an Imaris surface around the dorsal Is MN cell body (Figure A) and measured the volume included in this surface as an estimate of the dendritogenesis. Compared to controls, *draper* mutants did not show a significant difference (Figure B). However, we admit that this approach is not the best estimate of dendrite volume because dendrites from dorsal and ventral Is MNs, and Is MNs in different hemisegments, show overlap. A better measurement will require us to integrate a sparse labeling system (such as MCFO) into the *draper* mutant background, but we think that is beyond the scope of this manuscript.

A

B

3. The authors use a series of cell-type specific Draper knockdowns (glia, muscle, and both) to assess where Draper is required for cross-neuron plasticity. As the authors have UAS-Draper lines, a nice complementary experiment would be to perform cell-type specific rescue in the Draper mutant backgrounds (*Draper*^{MUT}, *repo*>UAS-Draper-I) to test if glial expression of Draper is sufficient to facilitate cross-neuron plasticity following ablation.

Reply: We thank the Reviewer for this suggestion, and we agree that a rescue experiment would bolster our findings. However, this rescue experiment also requires genetic components to ablate Is MNs. For example, in our overexpression system, the genotype of glial *draper-I* overexpressing animals is: *LexAOP-rpr /UAS-draper-I; Repo-GAL4/A8-LexA, LexAOP-GFP*. Putting all these constructs into a *draper* mutant will require several recombination and many months to just generate the line in addition to the electrophysiology experiments. Even without this rescue experiment, the *draper-I* overexpression experiments suggest that glia and/or muscle are able to facilitate cross-neuron plasticity.

Minor Comments:

1. As the authors switch their model neurons a couple of times in the manuscript, a small cartoon in each figure highlighting which muscle and MN are being assessed would be helpful.

Reply: We thank the Reviewer for this suggestion. We have inserted a cartoon schematic in the electrophysiology figures to show the muscle we recorded from, as well as a label in each image showing the muscle number.

2. Typo: Discussion, line 352, should read "their structural and functional"

Reply: We thank the Reviewer's keen eyes and have corrected this typo.

Overall, I commend the authors of their work. It's very well done, and merits publication in Nature Communications.

Reviewer #3 (Remarks to the Author):

Wang et al. investigate the signaling systems involved in what they refer to as "cross-neuron plasticity" using the *Drosophila* NMJ as a model system. Building off of previous work from their group and others, they find that genetic ablation of the *Is* motor neuron leads to both structural and functional changes in the remaining *Ib* motor neuron. Here, they test the hypothesis that Draper-mediated signaling is required for the cross-neuron plasticity observed, and find that indeed *Drpr* signaling in glia (and likely muscle as well) is necessary. Beyond this major finding, they also show that *drpr* overexpression can "boost" this synaptic plasticity, that functional plasticity can be induced throughout larval development (while only early stages of *Is* ablation can induce structural plasticity), and show some changes in locomotor activity.

The manuscript is clear, well written, and the experiments are in general performed at a high quality. The major finding, that *Drpr* signaling is necessary for cross-neuron plasticity, is of significant importance and impact to the field, as it reveals the first insights into how plasticity induced by ablation of one motor neuron is propagated to neighboring neurons. This result is not unexpected, given the seminal work by Marc Freeman showing that glia and *Drpr* signaling similarly propagate adaptations in bystander neurons in the wing after injury, but it is an important finding nonetheless. It remains unclear how *Drpr* operating in glia/muscle signals to neurons, and mechanistic insight into this would really add a lot to this manuscript. There are also some significant weaknesses in the areas discussed below that need to be addressed, particularly in the authors approach and analysis of synaptic function. Nonetheless, establishing *drpr*-mediated signaling and role of glia and muscles in transmitting cross-neuronal plasticity generates an exciting foundation to understand how healthy neurons adapt to neuronal death and will be of broad interest and impact to the field.

Major points:

1. Electrophysiological analysis: There are several significant issues with the electrophysiology presented in this manuscript.

1A: The authors use an indirect measure of *Ib/Is+Ib* to attempt to estimate EPSP amplitude coming from *Ib* and *Is*. It seems that throughout this manuscript, the authors should just report *Ib* EPSP amplitude, which is all they are really interested in. What is the point of showing the ratio between *Ib/Ib+Is*?

Reply: We really appreciate the Reviewer mentioning this point. The Ib/Ib+Is ratio is calculated from a direct measurement of Ib EPSP and Ib+Is EPSP. The ratio indicates the contribution of Ib EPSP in the combined Ib+Is EPSP. We completely agree that measuring Ib EPSP from all non-ablated conditions will be the best control. However, there are some technical caveats preventing us to directly measure Ib EPSP in conditions including mutant, knockdown or overexpression (see reply below).

The reason we show Ib/Ib+Is ratio in our normalized EPSP plot is to use this reference as a baseline Ib contribution. If Ib EPSP is not changed after Is MN ablation, then normalizing Ib EPSP to respective Ib+Is control should result in a similar value. However, if compensation happens, then we would expect a higher ratio than this baseline. Because our non-ablated condition (Ib+Is) for each experiment is carefully controlled, we think it is fair to compare the normalized EPSP to this ratio. Also, this Ib/Ib+Is baseline was used in our previous study (Wang et al., 2021) so a similar analysis would facilitate comparisons.

The functional plasticity the authors are interested in is just Ib at baseline and after Is ablation in all of the experiments presented. Hence, it would be clearest and most straightforward to just show Ib EPSP amplitudes and ignore the Ib+Is blended EPSP at baseline. For example, in Fig. 2F, only Ib EPSPs should be shown and made clear based on their simultaneous optical recordings.

Reply: (Figure 2F is current Fig. 2g) In addition to the comparison between “Ib at baseline” to “Ib activity after Is ablation”, another important comparison in this study is “Ib activity in different ablation conditions”. For example, a regular ablation versus ablation with *draper* knockdown.

This is assuming the authors can be absolutely sure that they are measuring just the Ib EPSP with their threshold manipulations. Ideally the key experiments would be done as in Fig. S1 visualizing the Ca response from Ib only to ensure they are really just measuring Ib EPSPs. Even better would be to use one of the more recent approaches the field has employed to selectively stimulate Is or Ib (optogenetics, botulinum toxin, etc).

Reply: We appreciate the Reviewer’s suggestion on different approaches to isolate Ib EPSP from the combined EPSP. However, we tried different approaches and encountered technical difficulties in teasing apart the Ib EPSP in our mutant, knockdown or overexpression conditions.

First, we used *MHC-GCaMP6f-Sh*. In experiments where we combine *MHC-GCaMP6f-sh* with electrophysiology (like Figure S1), there are two copies of *MHC-GCaMP6f-Sh*. However, in the mutant, knockdown or overexpression backgrounds, only one copy of *MHC-GCaMP6f-sh* was expressed due to the complex genetic background. We found that a single copy of *MHC-GCaMP6f-Sh* bleaches very fast and becomes indistinguishable from its high NMJ background (the high background is due to the *sharker* tag). In addition, our electrophysiology/imaging setup is comprised of an LED lamp and a widefield scope which need higher power to achieve high contrast. Other setups combining ephys with imaging typically including a spinning disk confocal which would allow for extended imaging with less bleaching (Newman et al., 2017). In addition, the higher Ca²⁺ concentration used in Newman et al., 2017 also increased contrast and lowered the bleaching.

Next, we expressed several versions of tagged *GCaMP8f* transgenes (generous gift from Dr. Dion Dickman) in Is MNs in our mutant, knockdown or overexpression system. We hypothesized that this would give a better signal to noise ratio and allow for extended imaging time since it is presynaptically tagged. However, we found that the morphology and physiology

of Is MNs were significantly altered when express GCaMP8f presynaptically, possibly due to our Is MN drive being too strong. We also tested a recently published *MHC-GCaMP8f* (Han et al., 2022), generous gift from Dr. Dion Dickman). This construct provided a much lower background and better Ca^{2+} dynamics. However, expressing this transgene alone decreased the EPSP significantly, at least for MN4-Ib. In addition, recombining *MHC-GCaMP8f* with *draper^{Δ5}* caused lethality.

Finally, we tested the recently published Botulinum neurotoxin (Han et al., 2022), generous gift from Dr. Dion Dickman). We expressed BoNT-C in Is MNs and expected to block both spontaneous and evoked activity from Is MNs. However, we found severely undergrown Is NMJs and even cell death of Is MNs (see Figure below), possibly due again to the strength of our Is MN driver. In images below, we show cell bodies of Is MNs expressing BoNT-C. Therefore, the mEPSP and EPSP we recorded from these animals are likely not from a “baseline” Ib MN because the Ib MNs may already compensate for these Is MN changes.

Is>BoNT-C

1B: Quantal content cannot be accurately compared between Is intact vs Is ablated. As the authors know, mEPSP amplitude is dramatically different (over 50%) between the two inputs due to differences in synaptic vesicle size and the glutamate receptor fields at each NMJ. Hence, even if the authors could accurately measure Ib EPSP amplitude (which it is not clear as discussed above), quantal content cannot be accurately determined by simply dividing the EPSP amplitude from the averaged mEPSP amplitude (which is a weighted average of minis coming from Is and Ib). While the quantal content value is not absolutely critical for the major findings in this manuscript, it is suggested that the authors explain this limitation in their approach and refer to their quantal content value as “apparent quantal content” or “estimated” quantal content.

Reply: We appreciate the Reviewer’s suggestions and have noted this in our Methods section (Line 529-531). We state that the quantal content calculated from the non-ablated condition is an estimated quantal content.

IC: It is strange that in Fig. S2, there is no change in mini amplitude or frequency with or without Is ablation. Do the authors have an explanation for this? From the Han et al. 2022 paper and others, it is very clear that minis are much larger at Is NMJs, so in Is ablated conditions mEPSP amplitude should be expected to be significantly smaller.

Reply: We thank the Reviewer for highlighting this insignificant change of mEPSP frequency and amplitude before and after Is ablation. Below, we will discuss them separately.

In wild type larvae, we agree that Is MNs have larger mEPSP amplitude than Ib MNs (several studies have reported this), and thus the average Ib mEPSP amplitude in Is ablated animals should be smaller. Indeed, we observed a trend of decrease. For example, in Figure S2b, $p=0.0672$ between control no Is ablation versus control Is ablation. Thus, we do not think there is compensation of the mEPSP amplitude at Ib NMJs. In addition, because we are comparing Ib only mEPSP amplitude with the weighted average of Ib+Is mEPSP amplitude, it is difficult to anticipate a significant difference without knowing the mEPSP frequency from each MN (weight). Therefore, the Ib mEPSP amplitude in Is ablated animals may decrease but it may not be significant. The mEPSP frequency of the Is MN on muscle 4 is indeed lower than MN4-Ib (Newman et al., 2017). Another explanation is that (Han et al., 2022) recorded from muscle 6 while we mainly recorded from muscle 4. Different Ib and Is MN combination may also cause a difference.

Similarly, there should be fewer minis released in the absence of Is, so mEPSP frequency should also be reduced.

Reply: For the mEPSP frequency, we do not anticipate a decrease, and we think there may be compensation of Ib mEPSP frequency (due to this non-decreased overall mEPSP frequency). This result is consistent with our previous findings (Wang et al., 2021): after Is ablation, both M4-Ib and M6-Ib showed a similar spontaneous frequency compared to non-ablated controls and MN12-Ib showed an even larger mEPSP frequency after Is ablation. However, we lack direct evidence to confirm the compensation of mEPSP frequency since we are unable to quantify the Ib mEPSP frequency in a wild-type condition. In addition, to avoid confusion, we did elaborate on this part in our manuscript.

There are also differences in the Ib EPSP amplitude compared to the Is contribution presented in this manuscript compared to previous studies. All of this was worked out in the Newman et al. 2017 and Han et al. 2022 studies. Do the authors think there is some functional plasticity induced in mini freq/amp that were not observed in the previous studies?

Reply: For the M4-Ib EPSP amplitude, in this study we revealed a 56% contribution to the Ib+Is EPSP, which is close to our previous observation (54.6%). In addition, the genetic background and Ca^{2+} concentrations used by different labs may also influence the EPSP contribution. For example, in the summary table below, we noticed large differences from studies that teased apart Ib EPSP and Is EPSP. In addition, (Newman et al., 2017) did not perform electrophysiology to distinguish Ib and Is EPSP.

Table Comparative Ib and Is physiology from prior NMJ studies.

muscle	Ib EPSP (mV)	Is EPSP (mV)	Ib+Is EPSP (mV)	Ib contribution	[Ca ²⁺]	reference
1	11.8	-	24.2	48.7%	0.3 mM	Aponte-Santiago et al., 2020
4	15.8	-	25.7	61.5%	0.5 mM	Wang et al., 2021
6	17.2	-	31.6	54.6%	0.5 mM	Wang et al., 2021
6	14.9	-	24.3	61.3%	1.0 mM	Li et al., 2002
6	~10	-	~35	28.6%	1.8 mM	Kurdyak et al., 1994
6	15.6	25.7	-	37.8%	0.5 mM	Genc et al., 2019*
6	15.6	-	23.1	67.5%	1.5 mM	Lnenicka and Keshishian, 2000
6	9.5	23.65	33.1	28.7%	0.5 mM	Han et al., 2022
7	12.1	-	15.5	43.8%	1.5 mM	Lnenicka and Keshishian, 2000
12	8.7	-	28.5	31.1%	0.5 mM	Wang et al., 2021

* This study used optogenetics, instead of voltage injection, to differentially activate Ib or Is.

2. Interpretation of Drpr overexpression results: The authors appear to really stress that Drpr is needed in glial cells to signal Is ablation and promote cross-neuron plasticity. But in most of their results, Drpr also appears to be needed in muscle as well. It is not clear why they emphasize Drpr being needed in glia so prominently.

Reply: We thank the Reviewer for mentioning the role of muscle in cross-neuron plasticity. In our *draper/shark* knockdown experiment, muscle knockdown did not eliminate structural plasticity or the elevated quantal content, compared to the complete elimination when knocking down in glial cells. Muscle knockdown did affect the normalized EPSP. The discrepancy between decreased EPSP and un-changed QC could be due to the slightly decreased mEPSP amplitude when ablating Is MNs (Figure S4d and S5d, as the reviewer mentioned above). Therefore, we think the muscle is not required for cross-neuron plasticity. In the revised manuscript, we further discuss the potential role of the muscle in the Discussion section (Line 398-406).

In addition, what does “enhanced plasticity” from Drpr overexpression actually mean?

Reply: “enhanced plasticity” means that the Ib MNs can compensate the EPSP/QC to a higher level when *draper-I* is overexpressed, compared to just Is ablation.

If the system wants to restore NMJ function to baseline (Is+Ib), shouldn't EPSP values from Ib be potentiated much more than what they observe? Even with drpr overexpression, EPSP amplitude from Ib is not sufficient to recover to baseline states (Is+Ib). So the reader is left wondering how to interpret these results;

Reply: We thank the Reviewer for bringing up this interesting point – to what extent cross-neuron plasticity can compensate. Unlike other plasticity mechanisms (such as PHP) where the healthy state (or baseline) is easier to measure, we cannot faithfully measure or anticipate the “real baseline” in our case. One important consideration here is the time of ablation. In most of our experiments, Is ablation happens in the late embryo, when both Ib and Is activity are lower. Therefore, the “baseline information” stored in the system is actually not the same level as the Ib+Is activity from the third instar. One piece of evidence that might support this hypothesis is that when we ablated Is MNs in later developmental stages, we observed a trend of more compensation (Figure 8i and 8j). The term “baseline” facilitates the understanding to a general audience.

what signal is actually being sensed.

Reply: Several Draper ligands have been found, and we mention them in the Discussion section (Line 372-378). For example, in the larval nervous system, a recent study from Chun Han's lab suggested that phagocytotic neurons expose phosphatidylserine as a ligand for Draper to instruct Wallerian degeneration (Ji et al., 2022). However, we cannot rule out additional unknown signaling mechanisms.

and why would Drpr levels be limiting in this pathway?

Reply: We think that the fact that the Ib MN does not compensate to wild-type Ib+Is level is not due to limited Draper levels, but instead, due to a lower "real baseline" compared to the wild-type third instar Ib+Is (similar to our response above). Alternatively, it is known that Draper-II inhibits Draper-I, and thus, Draper-I activity may be limited, or being regulated in a wild-type condition. We agree that understanding how Draper is regulated in healthy and degenerating conditions is an interesting question, but it is not the goal of this study.

What parameter is actually being targeted in this "cross neuron plasticity"?

Reply: We do not have direct evidence to support this hypothesis – but we anticipate that the release probability is increased at Ib NMJs. We observed both increased mEPSP frequency (ie. an un-changed overall mEPSP frequency) and increased quantal content in Is ablated larvae, which are both influenced by release probability. Alternatively, other synaptic parameters could also be regulated, such as active zone structure, active zone states, etc. We discussed these possibilities in the discussion section (Line 436-446). However, it is not straightforward to tease apart these synaptic parameters for the Ib MN alone. One approach might be focal macropatch recordings (Vasin and Bykhovskaia, 2017; Vasin et al., 2019) but this is beyond the scope of this study.

Most importantly, the authors move to the muscle 6 NMJ to do this experiment, and the rationale for this is not established. They mention their apparent concern that the muscle 4 NMJ may be "significantly higher...and may not show further enhancement" line 262. This possibility does not relieve them from doing and presenting the experiment at muscle 4 and, if they do not see any change in plasticity, they might then go to muscle 6. Indeed, it's not clear why they just ignored the muscle 4 NMJ that they did all of their previous experiments in. At minimum, all of the experiments shown in Fig. 4 should be first done and presented at the muscle 4 NMJ, and then they can also show the muscle 6 data. If Drpr overexpression has no impact on muscle 4 plasticity, that is an important result that needs to be shown up front.

Reply: We thank the Reviewer for mentioning this weakness in our manuscript. To address this issue, we performed recordings from *draper* overexpressing animals in this revised manuscript as suggested by the Reviewer. In short, we did not observe a further increase of plasticity at MN4-Ib when overexpressing *draper-I*, but we found a suppression of plasticity when overexpress *draper-II*. These results align with our expectations. We updated the overexpression section (Line 245-286) to include these experiments.

3. As the authors note there are two Is motor neurons that innervate separate sets of muscles. It would be of interest to manipulate both Is motor neurons and more deeply probe plasticity. In particular, the following experiments should be considered:

- At muscles innervated by the non-ablated Is input, does the Ib (or Is) motor neurons adapt or change when Is is ablated at the other muscle segments?

Reply: We thank the Reviewer for bringing up this interesting question. Our *Is-GAL4* expresses in both dorsal and ventral *Is* MNs, and our ablation paradigm efficiently ablates both at embryo stages (Wang et al., 2021). However, MN11-Ib, an Ib MN that does not receive *Is* co-innervation, shows plasticity when the *Is* MNs are ablated (Figure 7). We hope this at least partially addresses the Reviewer's question.

-What happens when both *Is* inputs are ablated? Does the Ib motor neuron show “enhanced cross neuron plasticity”, both in terms of structural and functional plasticity?

Reply: As discussed above, *Is>rpr,hid* ablates both MNs at 100% efficiency in the late embryo stage. Thus, the plasticity we observed is indeed the condition mentioned by the Reviewer. However, the Reviewer brings up an interesting point – does plasticity change based on the number of MNs ablated? This could also be extended to include sensory neuron ablation since their axons share nerves with MNs. However, we think this is beyond the scope of this study.

- How is larval locomotor behavior altered when both *Is* motor neurons are ablated (experiments like the ones presented in Fig. 7)?

Reply: (Figure 7 is current Figure 9) As mentioned above, the behavioral change we observed is caused by ablating both *Is* MNs (the *Is-GAL4* is expressed in both *Is* MNs).

4. Behavioral experiments presented in Fig. 7 – an absolutely crucial experiment to properly interpret the behavioral results is to perform the same experiments in baseline and *Is* ablated animals in a *Drpr* mutant background to determine whether the NMJ plasticity is actually responsible for the behavioral changes (as opposed to non-specific developmental or pre-motor circuit changes).

Reply: (Figure 7 is current Figure 9) The Reviewer is absolutely correct. To address this concern, we performed the locomotor assay in *draper* mutant larvae with or without *Is* ablation (Line 338-343). We found that the increased crawling speed is eliminated by *draper* mutation, but the turn frequency is not affected, suggesting that the plasticity is responsible for the behavioral changes. However, we found that *draper* mutant larvae crawl in an unsynchronized manner with a significant slower speed, which may mask the compensation from cross-neuron plasticity. Note that we also optimized our customized code to filter out the unsynchronized body contraction (which sometimes could be tracked as a “turn”). We re-processed the data from control and *Is* ablated animals using the updated code.

Minor points:

1. Some changes to the text need to be made:

Abstract line 40: “...axon terminal size and activity”.- This should be “axon terminal NUMBER and activity” if the authors are referring to the moderate increase in bouton number.

Reply: We agree that “axon terminal size” is ambiguous. We updated it into “terminal bouton number”. We think “axon terminal number” may be wrongly interpreted as the number of terminal branches.

Line 66: Should also cite PMID: 29748610

Reply: We appreciate the Reviewer reminding us of this relevant study. We have added this reference. We also included a missing reference PMID: 32345746 at appropriate location (Line 303).

Typos: Line 67 “the” should be “for”; line 79 remove “the”;

Reply: We thank the Reviewer pointing out these typos. We have corrected them in the revised manuscript.

2. In some areas of the manuscript, the authors use the term “homeostatic”, where it seems the term “adaptive” is more appropriate. Homeostatic refers to restoring fully baseline states, while the functional changes shown appear to enhance release, synaptic transmission is not restored to wild type baseline states (normal EPSP amplitudes in Is+Ib).

Reply: We agree with the Reviewer’s point – this plasticity change of Ib MNs is not homeostatic, but adaptive. We have removed the term “homeostasis” in Line 370.

Fuentes-Medel, Y., Logan, M. A., Ashley, J., Ataman, B., Budnik, V. and Freeman, M. R. (2009). Glia and Muscle Sculpt Neuromuscular Arbors by Engulfing Destabilized Synaptic Boutons and Shed Presynaptic Debris. *Plos Biol* 7, e1000184.

Goel, P., Khan, M., Howard, S., Kim, G., Kiragasi, B., Kikuma, K. and Dickman, D. (2019). A Screen for Synaptic Growth Mutants Reveals Mechanisms That Stabilize Synaptic Strength. *J Neurosci* 39, 4051–4065.

Han, Y., Chien, C., Goel, P., He, K., Pinales, C., Buser, C. and Dickman, D. (2022). Botulinum neurotoxin accurately separates tonic vs. phasic transmission and reveals heterosynaptic plasticity rules in *Drosophila*. *Elife* 11, e77924.

Ji, H., Sapar, M. L., Sarkar, A., Wang, B. and Han, C. (2022). Phagocytosis and self-destruction break down dendrites of *Drosophila* sensory neurons at distinct steps of Wallerian degeneration. *Proc National Acad Sci* 119, e2111818119.

Kerr, K. S., Fuentes-Medel, Y., Brewer, C., Barria, R., Ashley, J., Abruzzi, K. C., Sheehan, A., Tasdemir-Yilmaz, O. E., Freeman, M. R. and Budnik, V. (2014). Glial Wingless/Wnt Regulates Glutamate Receptor Clustering and Synaptic Physiology at the *Drosophila* Neuromuscular Junction. *J Neurosci* 34, 2910–2920.

Newman, Z. L., Hoagland, A., Aghi, K., Worden, K., Levy, S. L., Son, J. H., Lee, L. P. and Isacoff, E. Y. (2017). Input-Specific Plasticity and Homeostasis at the *Drosophila* Larval Neuromuscular Junction. *Neuron* 93, 1388-1404.e10.

Vasin, A. and Bykhovskaia, M. (2017). Focal Macropatch Recordings of Synaptic Currents from the *Drosophila* Larval Neuromuscular Junction. *J Vis Exp*.

Vasin, A., Sabeva, N., Torres, C., Phan, S., Bushong, E. A., Ellisman, M. H. and Bykhovskaia, M. (2019). Two Pathways for the Activity-Dependent Growth and Differentiation of Synaptic Boutons in *Drosophila*. *Eneuro* 6, ENEURO.0060-19.2019.

Wang, Y., Lobb-Rabe, M., Ashley, J., Anand, V. and Carrillo, R. A. (2021). Structural and functional synaptic plasticity induced by convergent synapse loss in the *Drosophila* neuromuscular circuit. *J Neurosci* JN-RM-1492-20.

REVIEWERS' COMMENTS

Reviewer #1 (Remarks to the Author):

In this revision manuscript, Wang et al. have refined their study about how cross-neuron plasticity works in *Drosophila* motor neurons. For the revision, the authors have augmented the work with new experiments and analyses. These include:

- new data adding draper mutants to the behavioral analysis to test the idea of cross-neuron plasticity
- new Draper-I overexpression data (showing no further increase in compensation)

as well as several additional levels of analysis done for specific queries from Reviewers 1-3.

The core findings are the same, and the quality of the work remains high. By characterizing a molecular mechanism involved in cross-neuron plasticity, the authors have extended important concepts of structural and functional plasticity for tonic and phasic motor neurons.

There are a couple of minor comments that are interpretation or stylistic. Note to editor: these comments are not extensive. They do not affect any of the core conclusions, and therefore they do not require any additional round of peer review.

Minor Comments

1. Lines 297-298: this experiment does not involve glia directly, so this reviewer would say that the result is consistent with the idea that glial cells play an important role (not "support our model").
2. Lines 324-325: I agree that this experiment shows that structural and functional plasticity are separable mechanistically. It is not the first place in the paper where that conclusion could be made. One could also make a similar conclusion from the Draper-I overexpression experiment (Lines 260-268).
3. Lines 269-277: This clarity of this paragraph could be improved with minor edits. First, the proposed Draper-II role for debris engulfment needs a reference. Second, the phenotypic concept jumps from debris engulfment to structural plasticity. Even though debris engulfment and structural plasticity are related ideas (Figs. 1-2), they are not the same thing, so the transition term "indeed" reads in a puzzling way. Finally, do the authors know if overexpressing Draper-II did anything to influence debris clearance at the NMJ?

Reviewer #2 (Remarks to the Author):

Congratulations to the authors on this very nicely revised manuscript! I have no further concerns.

Reviewer #3 (Remarks to the Author):

The authors have done a good job of responding to my concerns and those of the other reviewers in this revision. I appreciate all the work the authors did to clarify the electrophysiological analyses, and understand the technical issues that prevented them from employing the other approaches more fully. I do think that it would be easier for the readers to understand if the authors just showed their "estimated" Ib EPSP (which they deduce from their I_b/I_s+I_b ratio) so it is clear, but I will leave it up to the authors to consider.

It is also an interesting point the authors made regarding what the true "baseline" state of the NMJ is given that Is ablation happens early in development, so that compensation may be tuned for synaptic strength at an earlier developmental stage. I did not appreciate this idea in the first reading, and including a brief discussion about this in the manuscript I think would be beneficial for readers.

The additional controls and analyses included in this revision further strengthen and expand the impact of this important study.

REVIEWERS' COMMENTS

Reviewer #1 (Remarks to the Author):

In this revision manuscript, Wang et al. have refined their study about how cross-neuron plasticity works in *Drosophila* motor neurons. For the revision, the authors have augmented the work with new experiments and analyses. These include:

- new data adding draper mutants to the behavioral analysis to test the idea of cross-neuron plasticity
- new Draper-I overexpression data (showing no further increase in compensation)

as well as several additional levels of analysis done for specific queries from Reviewers 1-3.

The core findings are the same, and the quality of the work remains high. By characterizing a molecular mechanism involved in cross-neuron plasticity, the authors have extended important concepts of structural and functional plasticity for tonic and phasic motor neurons.

There are a couple of minor comments that are interpretation or stylistic. Note to editor: these comments are not extensive. They do not affect any of the core conclusions, and therefore they do not require any additional round of peer review.

Minor Comments

1. Lines 297-298: this experiment does not involve glia directly, so this reviewer would say that the result is consistent with the idea that glial cells play an important role (not “support our model”).

Reply: We appreciate this valuable suggestion, and we updated the phrase in Line 308.

2. Lines 324-325: I agree that this experiment shows that structural and functional plasticity are separable mechanistically. It is not the first place in the paper where that conclusion could be made. One could also make a similar conclusion from the Draper-I overexpression experiment (Lines 260-268).

Reply: We strongly agree with the Reviewer that there are several hints about separate mechanisms of structural and functional plasticity before acute ablation experiment. We mentioned this possibility at Line 277-278 in the final submission. We also mentioned the possibility that the lack of further increase of bouton number could also be due to the ceiling effect.

3. Lines 269-277: This clarity of this paragraph could be improved with minor edits. First, the proposed Draper-II role for debris engulfment needs a reference.

Reply: We added reference for the proposed role of Draper-II.

Second, the phenotypic concept jumps from debris engulfment to structural plasticity. Even though debris engulfment and structural plasticity are related ideas (Figs. 1-2), they are not the same thing, so the transition term “indeed” reads in a puzzling way.

Reply: We agree with the Reviewer that “indeed” is not appropriate in this context and we have removed it.

Finally, do the authors know if overexpressing Draper-II did anything to influence debris clearance at the NMJ?

Reply: We did not quantify this change but there appear to be more satellite boutons and ghost boutons when overexpressing Draper-II in the muscles, and bouton morphology sometimes appears abnormal (Supplementary Fig 9 e-f). However, *draper-II* is not expressed in body wall muscles (Fuentes-Medel et al., 2009). Therefore, we think that this could be gain-of-function phenotype.

Reviewer #2 (Remarks to the Author):

Congratulations to the authors on this very nicely revised manuscript! I have no further concerns.

Cheers!

Reviewer #3 (Remarks to the Author):

The authors have done a good job of responding to my concerns and those of the other reviewers in this revision. I appreciate all the work the authors did to clarify the electrophysiological analyses, and understand the technical issues that prevented them from employing the other approaches more fully. I do think that it would be easier for the readers to understand if the authors just showed their “estimated” Ib EPSP (which they deduce from their Ib/Is+Ib ratio) so it is clear, but I will leave it up to the authors to consider.

Reply: We agree with the Reviewer that showing “estimated” Ib EPSP could be easy to for understanding the comparison within a group (non-ablated versus ablated) to see if compensation happens upon ablation. However, such method does not support the comparison of ablated EPSP in different conditions (for example, Fig 3k and 3l), because the effects from genetic background will contribute. We do observe changes in Ib+Is EPSP when changing *draper* levels (for example, Supplementary Fig 9h). In addition, the graph will be busy since there will be eight columns. Therefore, we decided to normalize the EPSP first and make both comparisons together.

It is also an interesting point the authors made regarding what the true “baseline” state of the NMJ is given that Is ablation happens early in development, so that compensation may be tuned for synaptic strength at an earlier developmental stage. I did not appreciate this idea in the first reading, and including a brief discussion about this in the manuscript I think would be beneficial for readers.

Reply: We thank the Reviewer’s suggestion, and we added a paragraph in the Discussion section regarding the level of compensation (Line 457-465). We avoid the word “baseline” because this term is used before for the Ib baseline activity.

The additional controls and analyses included in this revision further strengthen and expand the impact of this important study.